# ON THE EXPRESSIVE POWER OF SPARSE GEOMETRIC MPNNS

**Yonatan Sverdlov**[1], **Nadav Dym**[1,2]

[1] Faculty of Mathematics
[2] Faculty of Computer Science
Technion – Israel Institute of Technology
`yonatans@campus.technion.ac.il`
`nadavdym@technion.ac.il`

## ABSTRACT

Motivated by applications in chemistry and other sciences, we study the expressive power of message-passing neural networks for geometric graphs, whose node features correspond to 3-dimensional positions. Recent work has shown that such models can separate *generic* pairs of non-isomorphic geometric graphs, though they may fail to separate some rare and complicated instances. However, these results assume a fully connected graph, where each node possesses complete knowledge of all other nodes. In contrast, often, in application, every node only possesses knowledge of a small number of nearest neighbors.

This paper shows that generic pairs of non-isomorphic geometric graphs can be separated by message-passing networks with rotation equivariant features as long as the underlying graph is connected. When only invariant intermediate features are allowed, generic separation is guaranteed for generically globally rigid graphs. We introduce a simple architecture, EGENNET, which achieves our theoretical guarantees and compares favorably with alternative architectures on synthetic and chemical benchmarks. Our code is available at GitHub.

## 1 INTRODUCTION

Geometric graphs are graphs whose nodes are a 'position' vector in $\mathbb{R}^d$ (typically $d = 3$) and whose symmetries include the permutation symmetries of combinatorial graphs and translation and rotation of the node positions. Geometric graphs arise naturally in learning applications for chemistry, physical dynamics, and computer vision as natural models for molecules, particle systems, and 3D point clouds. These applications motivated the introduction of many learning models for geometric graphs, which were often inspired by 'standard' *graph neural networks* (GNNs) for combinatorial graphs (Satorras et al., 2021; Han et al., 2024; Gasteiger et al., 2021). Subsequently, several theoretical works aimed primarily at understanding the expressive power and limitations of GNNs for geometric graphs.

GNNs for geometric graphs produce global graph features invariant to geometric graph symmetries. The expressive power of GNNs is typically assessed primarily by their ability to assign different global features to pairs of geometric graphs that are not geometrically isomorphic.

Recent research on this problem has uncovered several interesting results, mainly assuming that the graphs are full (an edge connects each pair of nodes). Under the full graph assumptions, several models for geometric GNNs are *complete*, that is, capable of separating *all* pairs of non-isomorphic graphs. However, this typically comes at a relatively high computational price. Examples of complete models include $(d-1)$-WL-based GNNs (Delle Rose et al., 2023; Hordan et al., 2023; Li et al., 2024a), GNNs based on sub-graph aggregation (Li et al., 2024b), and message passing models based on arbitrarily high-dimensional irreducible representations (Thomas et al., 2018; Gasteiger et al., 2021; Dym & Maron, 2020; Finkelshtein et al., 2022). This paper focuses on more efficient geometric GNNs: message-passing neural networks (MPNN) based on simple invariant and equivariant features. In (Pozdnyakov & Ceriotti, 2022; Li et al., 2024a), it was shown that, even under the full graph assumption, invariant MPNN-based networks are not complete. In contrast, Li et al. (2024b) and

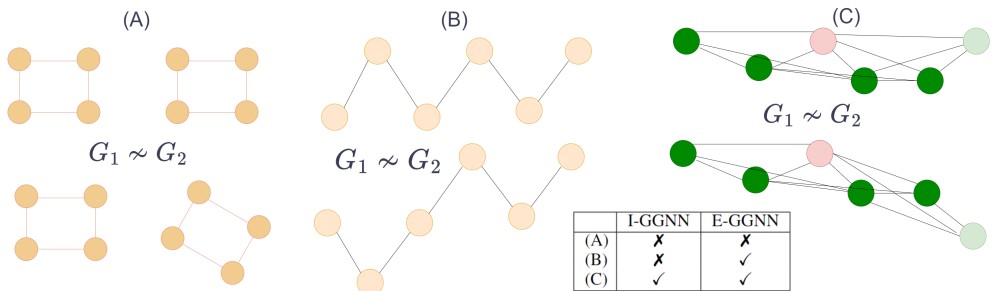

Figure 1: (A) and (B) are pairs of non-isomorphic geometric graphs with the same distances along edges. I-GGNN cannot distinguish such pairs, while E-GGNN can (generically) if the graph is connected, as in (B). Subplot (C) depicts an example of globally rigid graphs, where non-isomorphic geometric graphs do not share the same distances across edges (in the figure, the edge between pink and light green). Such examples can be separated by both I-GGNN and E-GGNN.

Hordan et al. (2023) proved that these models are 'generically complete,' which means that, under the full graph assumption, they are capable of separating most geometric graphs and only fail on a Lebesgue measure zero set of corner cases, which are difficult to separate.

In practical applications, the full graph assumption is often not fulfilled: to improve computational efficiency, geometric GNNs often operate on sparse graphs, where each node is only connected to a few nearest neighbors. Accordingly, the first question we address in this paper is:

**Question 1.** For which graphs, other than the full graphs, are geometric message-passing neural networks generically complete?

Our answer to this question is inspired by the results of Joshi et al. (2023). They divided Message-passing-based networks for geometric graphs into *Equivariant Geometric Graph Neural Networks* (E-GGNNs), which use rotation *equivariant* hidden features, and *Invariant Geometric Graph Neural Networks* (I-GGNNs), which use rotation *invariant* hidden features, and showed that E-GGNNs have substantial advantages over I-GGNNs in terms of their expressive power.

Our answer to Question 1 is also strongly related to this dichotomy. We prove in Theorem 4.1 that E-GGNNs are generically complete for a given graph if and only if the graph is connected. This is visualized in Figure 1. E-GGNNs will not be able to differentiate between the pair of geometric graphs in Figure 1(A) as the underlying graph is disconnected but will be able to differentiate between the pairs in (B) and (C). In contrast, I-GGNNs will not be able to differentiate between the non-isomorphic pairs in Figure 1(A)-(B), as the distances across all graph edges are the same. On the other hand, in (C), the distances along graph edges determine the positions of all nodes up to rotation and translation. In rigidity theory, such graphs are called 'generically globally rigid.' In Theorem 3.1, we prove that I-GGNNs are generically complete if and only if the underlying graph is generically globally rigid. These two theorems give a complete answer to Question 1.

To guarantee generic separation, our theorems make the standard assumption (Joshi et al., 2023) that the I-GGNN and E-GGNN we consider are maximally expressive, meaning that all aggregation and readout functions used throughout the networks are injective. Recent work has shown that relatively standard and simple I-GGNN can fulfill this assumption (Amir et al., 2023). However, for E-GGNN, it is unclear how this assumption can be fulfilled. Accordingly, our second question is

**Question 2.** Can we devise a simple E-GGNN architecture that is maximally expressive hence, generically complete on connected graphs?

We answer this question affirmatively by devising a simple architecture, which we name EGENNET, and proving in Theorem 5.1 that it is maximally expressive (for generic graphs). This architecture is based on a simple message-passing architecture that resembles EGNN (Satorras et al., 2021) but uses multiple equivariant channels as in (Levy et al., 2023).

We experimentally find that EGENNET is highly successful in synthetic separation tasks and on several learning tasks involving medium-sized molecules, with a performance comparable to, or better than, state-of-the-art methods. These results indicate that simple, generically complete methods

like EGENNET are sufficient for the tasks we considered. However, we acknowledge that there may be other tasks, perhaps involving molecules with non-trivial symmetry groups, where more complex geometric graph neural networks that offer full separation could be beneficial.

**Contributions**   To summarize, our main results are

- Showing generic separation of E-GGNNs depends on graph connectivity.
- Showing generic separation of I-GGNNs depends on graph rigidity.
- Proposing a simple E-GGNN architecture that is provably (generically) maximally expressive and performs well in practical tasks.

**Appendix**   See the appendix for all full proofs and related work.

## 2   SETUP: GEOMETRIC GRAPH NEURAL NETWORKS AND RIGIDITY THEORY

**Geometric Graphs.**   In our discussion of geometric graphs and GNNs, we loosely follow the definitions of Joshi et al. (2023). We define a geometric graph as a pair $\mathcal{G} = (\boldsymbol{A}, \boldsymbol{X})$. The matrix $\boldsymbol{A} \in \{0,1\}^{n \times n}$ is an adjacency matrix, which fully defines a combinatorial graph. The neighborhood of a node $i$ is denoted by $\mathcal{N}_i$. Accordingly, we sometimes use statements like 'the graph $\boldsymbol{A}$.' We assume for simplicity that the graph nodes are the integers $1, \ldots, n$. The matrix $\boldsymbol{X} = (\boldsymbol{x}_1, \ldots, \boldsymbol{x}_n) \in \mathbb{R}^{n \times d}$ denotes node positions. We denote the set of all geometric graphs with coordinates in $\mathbb{R}^d$ by $\mathbb{G}(d)$, and the subset of graphs with $\leq N$ nodes by $\mathbb{G}_N(d)$. In Appendix H, we discuss incorporating additional node and edge features into our model and theoretical results.

The symmetry group of geometric graphs is the product of the permutation group $S_n$ and the group of rigid motions $\mathcal{E}(d)$. The action of a permutation matrix $\boldsymbol{P}$ on a geometric graph is given by $P\mathcal{G} := (\boldsymbol{P}\boldsymbol{A}\boldsymbol{P}^\top, \boldsymbol{P}\boldsymbol{X})$. A rigid motion in $\mathcal{E}(d)$ is a rotation $\boldsymbol{Q} \in \mathcal{O}(d)$ and a translation vector $\boldsymbol{t} \in \mathbb{R}^d$. They act on $\boldsymbol{X}$ by applying the rotation and translation to all its coordinates, $\boldsymbol{x}_i \mapsto \boldsymbol{Q}\boldsymbol{x}_i + \boldsymbol{t}$.

Two geometric graphs $\mathcal{G}$ and $\mathcal{H}$ are *geometrically isomorphic* if there exists a transformation in $S_n \times \mathcal{E}(d)$ which can be applied to $\mathcal{G}$ to obtain $\mathcal{H}$. We will say that a function $F$ defined on $\mathbb{G}(d)$ is *invariant* if $F(\mathcal{G}) = F(\mathcal{H})$ for all $\mathcal{G}$ and $\mathcal{H}$ which are geometrically isomorphic.

The expressive power of an invariant model $F$ is closely related to its ability to separate pairs $\mathcal{G}, \mathcal{H}$ that are not geometrically isomorphic: clearly, if $F$ cannot separate $\mathcal{G}$ from $\mathcal{H}$, then $F$ will not be a good approximation for functions $f$ for which $f(\mathcal{G}) \neq f(\mathcal{H})$. Conversely, if $F$ can separate all pairs of geometric graphs, then it can be used to approximate all continuous invariant (Dym & Gortler, 2024; Hordan et al., 2023) and equivariant (Villar et al., 2021; Hordan et al., 2024) functions. In Appendix F, we discuss the connection between approximation and separation in the context of our results.

Constructing an $F$ which can separate all geometric graphs is at least as challenging as distinguishing all combinatorial graphs (by setting all node features to zero). Without any assumptions on the point cloud, achieving this is infeasible. To address this, we employ the concept of generic geometric graphs which originates from rigidity theory.

**Generic separation and Rigidity Theory**   As discussed in the introduction, our paper focuses on generic separation. The results we present in this section are primarily influenced by rigidity theory, so we also use this field's common notion of 'generic'. We will say that $\boldsymbol{X} \in \mathbb{R}^{d \times n}$ is *generic* if it is not the root of any multivariate polynomial $p : \mathbb{R}^{d \times n} \to \mathbb{R}$ with rational coefficients. The genericity assumption guarantees that $\boldsymbol{X}$ avoids many degeneracies, making separation easier. For example, for generic $\boldsymbol{X}$ all pairs of points have different norms, and every $d$ tuple of points $\boldsymbol{x}_{i_1}, \ldots, \boldsymbol{x}_{i_d}$ is full rank because $\|\boldsymbol{x}_i\|^2 - \|\boldsymbol{x}_j\|^2$ and $\det[\boldsymbol{x}_{i_1}, \ldots, \boldsymbol{x}_{i_d}]$ are polynomials with natural coefficients. We note that Lebesgue almost all point clouds in $\mathbb{R}^{d \times n}$ are generic (Mityagin, 2020).

A globally rigid graph is a central topic in rigidity theory:

**Definition 2.1** (Globally rigid). ( Garamvölgyi et al. (2022)) A geometric graph $\mathcal{G} = (\boldsymbol{A}, \boldsymbol{X})$ is *globally rigid* in $\mathbb{R}^d$, if the only $\boldsymbol{X}' \in \mathbb{R}^{n \times d}$ which satisfy that $\|\boldsymbol{x}_i - \boldsymbol{x}_j\| = \|\boldsymbol{x}_i' - \boldsymbol{x}_j'\|$ for all edges $(i, j)$ in $\boldsymbol{A}$, are those that are related to $\boldsymbol{X}$ by a rigid motion.

A combinatorial graph $\boldsymbol{A}$ is *generically globally rigid* in $\mathbb{R}^d$, if the geometric graph $\mathcal{G} = (\boldsymbol{A}, \boldsymbol{X})$ is *globally rigid* in $\mathbb{R}^d$ for every generic $\boldsymbol{X} \in \mathbb{R}^{n \times d}$.

The geometric graphs in Figure 1 (A)-(B) are not globally rigid, as they have the same distances along edges but are not related by a rigid motion. The underlying combinatorial graph, disconnected in (A) or line graph in (B), is not generically globally rigid. In contrast, the underlying combinatorial graph in (C) is generically globally rigid. Another simple example of a generically globally rigid graph is the full graph. Additional non-trivial examples can be found in (Jordán & Tanigawa, 2022; Cheung & Whiteley, 2005). A necessary condition for generic global rigidity of a graph with $n \geq d + 2$ nodes in $\mathbb{R}^d$ is to be $d + 1$ connected [*], which explains why the graphs in (A)-(B) are not generically globally rigid.

Note that global rigidity definitions focus on reconstructing $\boldsymbol{X}$ when $\boldsymbol{A}$ is known. In the study of GGNNs, we are interested in identifying both $\boldsymbol{X}$ and $\boldsymbol{A}$:

**Definition 2.2.** Let $d$ be a natural number. Following Li et al. (2024b), we will say that an invariant function $F$ defined on $\mathbb{G}(d)$, *identifies* a geometric graph $\mathcal{G}$, if for every $\hat{\mathcal{G}} \in \mathbb{G}(d)$, we have that $F(\mathcal{G}) = F(\hat{\mathcal{G}})$ if and only if $\mathcal{G}$ and $\hat{\mathcal{G}}$ are geometrically isomorphic.
We say that $F$ *generically identifies* a *combinatorial graph* $\boldsymbol{A}$ in $\mathbb{R}^d$, if $F$ can identify $\mathcal{G} = (\boldsymbol{A}, \boldsymbol{X})$ for every generic $\boldsymbol{X} \in \mathbb{R}^{n \times d}$.
We say that $F$ *generically fails to identify* a *combinatorial graph* $\boldsymbol{A}$ in $\mathbb{R}^d$, if $F$ does not identify $\mathcal{G} = (\boldsymbol{A}, \boldsymbol{X})$ for every generic $\boldsymbol{X} \in \mathbb{R}^{n \times d}$.

Note that the fact that $F$ does not generically identify $\boldsymbol{A}$ only guarantees that there exists a generic $\boldsymbol{X}$ so that $F$ fails to identify $(\boldsymbol{A}, \boldsymbol{X})$. The statement '$F$ generically fails to identify' is, apriori, significantly stronger. Also note that invariant $F$ separates all non-isomorphic graphs (is *complete*) if and only if it identifies all graphs. As discussed in the introduction, our goal will be to classify which graphs are generically identifiable by standard message-passing-based networks for geometric graphs, which can be classified as either I-GGNN or E-GGNN. We will now define these concepts.

**E-GGNN** In the context of combinatorial graphs (with no geometric information), Gilmer et al. (2017); Jogl et al. (2023) and Veličković (2022) showed that many famous graph neural networks are instantiations of MPNNs. The goal of Joshi et al. (2023) was to show, analogously, that many neural networks for *geometric* graphs follow a generalized MPNN framework. Loosely following Joshi et al. (2023), we define *Equivariant Geometric Graph Neural Networks (E-GGNN)* to be functions defined by a sequence of layers that propagates vector features from iteration $t$ to $t + 1$ via a learnable function $f_{(t)}$ (at initialization we set $v_j^{(0)} = 0$).

$$\boldsymbol{v}_i^{(t+1)} = f_{(t)} \left( \{\!\{ \boldsymbol{v}_j^{(t)}, \boldsymbol{x}_i - \boldsymbol{x}_j, \mid j \in \mathcal{N}_i \}\!\} \right). \tag{1}$$

To ensure the equivariance of the construction, the function $f_{(t)}$ is required to be rotation equivariant. Translation invariance is implicitly guaranteed by the translation invariance of $\boldsymbol{x}_i - \boldsymbol{x}_j$, and the requirement that $f_{(t)}$ is defined on multi-sets implicitly enforces permutation equivariance. We will often refer to $f_{(t)}$ as an *aggregation function*.

We note that for the sake of simplicity, we don't require a *combine* step that incorporates $\boldsymbol{v}_i^{(t)}$ in the process of computing $\boldsymbol{v}_i^{(t+1)}$ since we find that this step has little influence on our theoretical and empirical results. We also note that the vector $\boldsymbol{v}_i^{(t)}$ is not necessarily $d$ dimensional, and the action of $O(d)$ on $\boldsymbol{v}_i^{(t)}$ is allowed to change from layer to layer. For example, in the EGENNET architecture we will introduce later on, each $\boldsymbol{v}_i^{(t)}$ will be a $d \times m$ matrix, and the action of $O(d)$ will be multiplication on each of the $m$ coordinates.

After $T$ iterations of E-GGNNs, a permutation, rotation, and translation *invariant* feature vector can be obtained via a two-step process involving a rotation-invariant function $f_{\mathrm{inv}}$ and a multi-set READOUT function:

$$\boldsymbol{s}_i = f_{\mathrm{inv}}(\boldsymbol{v}_i^{(T)}), \quad s^{\mathrm{global}} = \text{READOUT} \{\!\{ \boldsymbol{s}_1, .., \boldsymbol{s}_n \}\!\} \tag{2}$$

---

[*]We say a graph $G$ is $k$ connected if the graph remains connected after removing any $k$ vertices.

As shown by Joshi et al. (2023), many popular geometric graph neural networks can be seen as instances of E-GGNNs with a specific choice of functions $f_{(t)}$ and readout functions. Examples include EGNN (Satorras et al., 2021), TFN (Thomas et al., 2018), GVP (Jing et al., 2020), and MACE (Batatia et al., 2022). In Section 5.1, we present EGENNET, our instantiation of this framework.

**I-GGNNs**   An important subset of E-GGNNs are *invariant* GGNNs (I-GGNNs). These networks only maintain scalar features $s_i^{(t)}$ and replace $\boldsymbol{x}_i - \boldsymbol{x}_j$ with its norm to obtain

$$s_i^{(t+1)} = f_{(t)}\left(\{\!\{s_j^{(t)}, \|\boldsymbol{x}_i - \boldsymbol{x}_j\|, \mid j \in \mathcal{N}_i\}\!\}\right) \tag{3}$$

A global invariant feature is obtained via a permutation invariant readout function as in equation 2. An advantage of I-GGNNs is that there are no rotation equivariance constraints on $f_{(t)}$. Therefore, any standard message-passing neural network that can process continuous edge weights can be employed as an I-GGNN by setting the edge weights to $\|\boldsymbol{x}_{ij}\|$. Examples of I-GGNNs in the literature include a variant of EGNN used for invariant tasks (Satorras et al., 2021), SchNet (Schütt et al., 2018), DimeNet (Gasteiger et al., 2020b) and SphereNet (Liu et al., 2021)[†]. Our definition of I-GGNN follows the definition from (Pozdnyakov & Ceriotti, 2022; Li et al., 2024a). In Appendix E, we explain that the definition of I-GGNN Joshi et al. (2023) has a seemingly minor difference that renders it significantly stronger.

**Maximally expressive E-GGNN and I-GGNNs**   The separation power of E-GGNN and I-GGNN architectures depends on the specific instance used. For example, if we choose the final readout function as the zero function, all geometric graphs will be assigned the same value. To address this, it is customary to assume that all functions $f_{(t)}$, as well as $f_{\text{inv}}$ and the readout function, are injective, up to the ambiguities these functions have by construction. Following Joshi et al. (2023), we will call E-GGNN (respectively I-GGNN) architectures, which are composed of injective functions, maximally expressive E-GGNN (respectively I-GGNN) architectures. We note that the existence of the injective functions necessary for maximal expressivity follows from set-theoretic considerations, similar to those used by Wang et al. (2024). The question of whether practical differentiable aggregation functions are injective is discussed in Section 5.

## 3   EXPRESSIVE POWER OF I-GGNN

Our next goal is to analyze the separation abilities of I-GGNN.

**Theorem 3.1.** *[expressive power of I-GGNN] Let $d$ be a natural number. Let $F$ be an I-GGNN. Let $A$ be a graph not generically globally rigid on $\mathbb{R}^d$. Then, $F$ generically fails to identify $A$. Conversely, if $A$ is generically globally rigid on $\mathbb{R}^d$ and $F$ is a maximally expressive I-GGNN with depth $T = 1$, then $F$ generically identifies $A$.*

*Proof idea.* if $A$ is not globally rigid, then according to Gortler et al. (2010), for *every* generic $X$ there exists $X'$ for which distances along edges are preserved, but $X$, $X'$ are not related by a rigid motion. It's easy to see any I-GGNN, which is based only on distances along edges, can't separate such graphs.

On the other hand, generic geometric graphs have distinct pairwise distances. The full proof (see Appendix G) shows that this can be used to reconstruct the combinatorial graph $A$. By global rigidity, we can then reconstruct $X$, up to rigid motion.   □

Note that Theorem 3.1 is a generalization of Hordan et al. (2023) that showed maximally expressive I-GGNN generically identify full graphs. We now showed this is true for all generically globally rigid graphs.

---

[†] DimeNet (Gasteiger et al., 2020b) and SphereNet (Liu et al., 2021) have access to the power graph $\mathcal{G}^2$ (defined later in the paper) and thus are invariant models on $\mathcal{G}^2$ and have potential expressive power stronger than invariant I-GGNN on $\mathcal{G}$.

**I-GGNN generic separation by power graph preprocessing**  A natural strategy to overcome the lack of generic separation for graphs that are not generically globally rigid, as proven in Theorem 3.1, is to replace these graphs with graphs that *are* generically globally rigid. This replacement can be done using power graphs: For a given graph $\mathcal{G}$ and natural number $k$, the power graph $\mathcal{G}^k$ is defined to have the same nodes as the original graph $\mathcal{G}$. A pair of nodes are connected by an edge in the power graph if and only if there is a path between them in the original graph of length $\leq k$.

In (Jordán & Tanigawa, 2022; Cheung & Whiteley, 2005), it was shown that if the original graph $\mathcal{G}$ is connected, then the power graph $\mathcal{G}^{d+1}$ is generically globally rigid in $\mathbb{R}^d$. Accordingly, when the input graph $\mathcal{G}$ is connected but not generically globally rigid, our theory suggests that applying I-GGNN not to the original graph but to its $d + 1$ power may benefit separation. An experiment illustrating this idea is presented in Table 1.

## 4   EXPRESSIVE POWER OF E-GGNN

When using E-GGNN rather than I-GGNN, we will have generic separation if and only if the graph is connected.

**Theorem 4.1.**  *[expressive power of E-GGNN] Let $d$ be a natural number. Let $F$ be an E-GGNN. Let $A$ be a disconnected graph. Then, $F$ generically fails to identify $A$.*
*Conversely, if $A$ is connected and $F$ is a maximally expressive E-GGNN with depth $T \geq d + 1$, then $F$ generically identifies $A$.*

*Proof idea.*  Suppose a geometric graph $\mathcal{G}$ isn't connected. We can apply two distinct rigid motions to two connected components and obtain a new geometric graph, $\mathcal{G}'$, which cannot be separated from $\mathcal{G}$ using E-GGNN. This is illustrated in Figure 1(A). Conversely, in the appendix Theorem G, we show that the features $s_i$ obtained after $d + 1$ iterations of a maximally expressive E-GGNN can encode the multi-set of all distances $\|x_i - x_j\|$ for all $j$ which are $d + 1$-hop neighbors of $i$ (a similar claim was shown by Joshi et al. (2023)). Thus, $d + 1$ iterations of maximally expressive E-GGNN can simulate the application of a single iteration of an I-GGNN on the $d + 1$ power graph. The proof then follows from the fact that the $(d + 1)$ power graph is generically globally rigid (Jordán & Tanigawa, 2022; Cheung & Whiteley, 2005).  □

An immediate result of this theorem is that generic graph separation is a local problem with a problem radius of $d + 1$ (Alon & Yahav, 2020) .[‡] We note that the generic assumption allows us to reconstruct the coordinates of the nodes and the combinatorial graph. Without the generic assumption, even models based on 2-WL that are complete on full graphs cannot reconstruct the combinatorial graph. Setting all positions to zero nullifies the geometric information, and 2-WL is insufficient to distinguish all combinatorial graphs.

## 5   BUILDING A GENERICALLY MAXIMALLY EXPRESSIVE E-GGNN

To obtain generic separation guarantees as in Theorems 3.1, 4.1 for a practical instantiation of E-GGNN (or I-GGNN), we will need to show that the instantiation is fully expressive. This type of question was studied rather extensively for combinatorial graphs (Amir et al., 2023; Aamand et al., 2022), and these results can be used to show that some simple realizations of I-GGNN are maximally expressive. We discuss this in more detail in Appendix C.

For E-GGNN, the requirement that $f_{(t)}$ is simultaneously injective (up to permutation) and equivariant to rotations seems to be challenging to attain with a simple E-GGNN construction, and indeed, Joshi et al. (2023) who suggested the notions of maximally expressive E-GGNN, did not give any indication of how they could be constructed in practice. Moreover, in Appendix D, we show that a maximally expressive E-GGNN can separate all pairs of full graphs and separate geometrically all pairs of connected graphs[§], indicating constructing such a network may be computationally expensive, as

---

[‡]A task's problem radius denoted by $r$ is the minimal integer where the task becomes solvable upon each node observing its neighborhood within a $r$-hop radius and aggregating all node's information.

[§]We define the geometric separation of a geometric graph if we can separate the point clouds up to group action, without necessarily reconstructing the combinatorial graph.

currently, such guarantees are only achieved by networks of relatively high complexity based on 2-WL subgraph aggregation techniques.

In our setting, however, the problem is less severe since, from the outset, we are only interested in analyzing network behavior on generic graphs. Accordingly, we would like to require our E-GGNN to be maximally expressive only for generic graphs. It turns out that this goal can be achieved with a straightforward E-GGNN architecture, which we name EGENNET. Our next step will be to present this architecture.

## 5.1 THE EGENNET ARCHITECTURE

Our goal in this section is to build a maximally expressive EGNN. For this, we need to implement injective readout and aggregation functions $f_{(t)}, f_{\text{inv}}, \text{READOUT}$ (see equations 1-2. The EGENNET architecture we suggest resembles an EGNN architecture (Satorras et al., 2021) with multiple equivariant channels as in (Levy et al., 2023). The EGENNET architecture depends on two hyper-parameters: depth $T$ (the term 'depth' refers to the number of MPNN iterations) and channel number $C$. The input to the architecture is a geometric graph $(\boldsymbol{A}, \boldsymbol{X})$.

**Aggregation** The architectures then maintains, at each iteration $t$ and node $i$, an 'equivariant' node feature with $C$ channels

$$\boldsymbol{v}_i^{(t)} = (v_{i,1}^{(t)}, \ldots, v_{i,C}^{(t)}), \quad v_{i,c}^{(t)} \in \mathbb{R}^d, c = 1, \ldots, C \tag{4}$$

These feature values are initialized to zero and are updated via the rotation and permutation equivariant aggregation formula

$$v_{i,q}^{(t+1)} = \sum_{j \in \mathcal{N}_i} \left( \phi^{(t,q,0)}(\|\boldsymbol{x}_i - \boldsymbol{x}_j\|, \|\boldsymbol{v}_j\|)(\boldsymbol{x}_i - \boldsymbol{x}_j) + \sum_{c=1}^{C} \phi^{(t,q,c)}(\|\boldsymbol{x}_i - \boldsymbol{x}_j\|, \|\boldsymbol{v}_j\|)v_{jc}^{(t)} \right) \tag{5}$$

where $\|\boldsymbol{v}_j\|$ is the $C$ dimensional vector containing the norms of the vectors $v_{jc}, c = 1, \ldots, C$.

**Construction of $f_{\text{inv}}$ and READOUT** After $T$ iterations, we obtain rotation and permutation invariant node features $\boldsymbol{s}_i \in \mathbb{R}^C$ and global feature $s^{\text{global}} \in \mathbb{R}^C$ via

$$s_{i,q} = \|\sum_{c=1}^{C} \theta_{c,q} v_{i,c}^{(T)}\|^2, \quad s_q^{\text{global}} = \sum_{i=1}^{n} \phi_{\text{global}}^{(q)}(\boldsymbol{s}_i), \quad q = 1, \ldots, C \tag{6}$$

Where $\theta$ is a learnable matrix, and the functions $\phi_{\text{global}}^{(q)}, \phi^{(t,q,c)}$ are all fully connected neural networks with a single layer, an output dimension of 1, and an analytic non-polynomial activation (in our implementation, TanH activation). The complexity of EGENNET is linear in $N \cdot C$, as we discuss in Appendix G.2.

The next theorem shows that EGENNET is generically maximally expressive:

**Theorem 5.1.** *Let $d, N, T$ be natural numbers. Let $F$ be a maximally expressive E-GGNN of depth $T$, and let $F_\theta$ denote the EGENNET architecture with depth $T$ and $C = 2Nd + 1$ channels. Then, for Lebesgue almost every choice of network parameters $\theta$, we have that for all generic $\mathcal{G}, \hat{\mathcal{G}} \in \mathbb{G}_N(d)$,*

$$F(\mathcal{G}) = F(\hat{\mathcal{G}}) \iff F_\theta(\mathcal{G}) = F_\theta(\hat{\mathcal{G}})$$

*Proof idea.* The proof relies on the methodology developed by Dym & Gortler (2024); Amir et al. (2023) and Hordan et al. (2023). We note that each step in EGENNET employs $C$ identical analytic functions indexed by $q = 1, \ldots, C$, with independent parameters. Dym & Gortler (2024) and Amir et al. (2023) showed that in this setting, if every pair of non-isomorphic inputs can be separated by some parameter of the original function, then $C = 2Nd + 1$ copies of this function with random parameters will suffice to separate *all pairs* uniformly. The proof is then reduced to showing that any pair of non-isomorphic inputs can be separated by some parameter vector, which we prove in the appendix G.1. □

Joining this theorem with our theorem on maximally expressive E-GGNN, we deduce

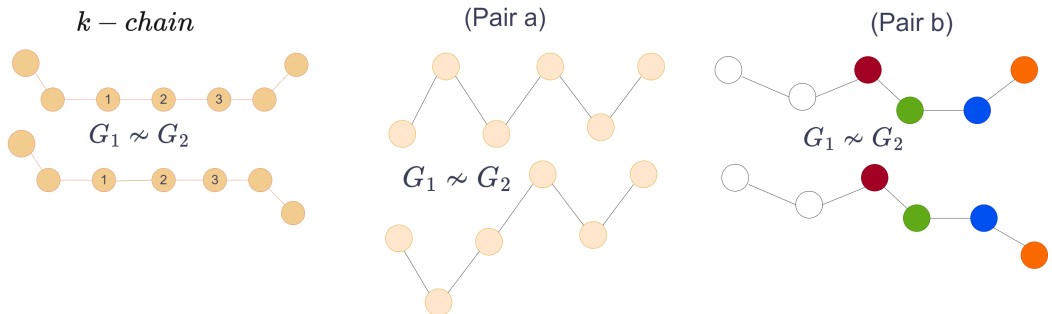

Figure 2: From Left to Right: $k$-chain pair graphs that are non-generic and require $\sim k/2$ blocks for separation, Pair a in which 2-power graph is enough for I-GGNN to distinguish and Pair b in which 3-power graph is necessary and sufficient for separation.

**Corollary 5.2.** *Let $d, N$ be natural numbers. let $F_\theta$ denote the* EGENNET *architecture with depth $T = d + 1$ and $C = 2Nd + 1$ channels. Then for Lebesgue almost every $\theta$, for every pair of graphs $\mathcal{G}, \mathcal{G}' \in \mathbb{G}_N(d)$ which are generic and connected,*

$$F_\theta(\mathcal{G}) = F_\theta(\hat{\mathcal{G}}) \iff \mathcal{G}, \hat{\mathcal{G}} \text{ are geometrically isomorphic}$$

Note that by Joshi et al. (2023), theorems 20, 22, the dimension of any injective embedding must be at least $nd - \frac{d(d-1)}{2}$, showing our embedding dimension $2nd + 1$ is optimal up to factor of 2 (when $n \gg d$).

# 6 EXPERIMENTS

In the experiments section, we have three main objectives. First, we aim to demonstrate that invariant models are indeed strictly less expressive than equivariant models and that power graphs help to mitigate this gap. Second, we seek to show that our model can distinguish challenging graph examples, including those requiring multiple layers for separation, thereby avoiding the bottleneck phenomenon encountered in other geometric models. Finally, we aim to demonstrate the improved performance of our model on real-world benchmarks.

## 6.1 SEPARATION EXPERIMENTS

**Enhancing Model Expressiveness: Power Graphs in Invariant vs. Equivariant Comparisons** We begin with experiments to corroborate our theoretical findings. We consider examples where the graph $\boldsymbol{A}$ is a line graph, which is connected but not generically globally rigid. We choose two pairs of geometric line graphs, Pair a and Pair b, depicted in Figure 2. These pairs have identical distances along graph edges but are not isomorphic. Therefore, they cannot be distinguished by any I-GGNN but (at least when generic) can be distinguished by E-GGNN and our EGENNET architectures. Following the protocol from Joshi et al. (2023), we construct binary classification problems from Pair a and Pair b and apply our E-GGNN architecture EGENNET, as well as three popular I-GGNN architectures: SchNet (Schütt et al., 2018), DimeNet (Gasteiger et al., 2020b), and SphereNet (Liu et al., 2021).

The results are shown in the first column in Table 1. As expected, EGENNET perfectly separates pairs a and b, while SchNet fails to separate both. Note that DimeNet and SphereNet separate Pair a but not Pair b. This is because DimeNet and SphereNet compute angles at edges, which utilizes information not contained in the graph $\mathcal{G}$ but only in the power graph $\mathcal{G}^2$. Indeed, distances between two-hop neighbors are sufficient for discriminating between the two geometric graphs in Pair a but not in Pair b, where the distance between the three-hop red and orange nodes is needed for separation. Next, as we suggested in Section 3, we apply the same algorithms to the powers of the original graph. As shown in Table 1, this improves the separation capabilities of the I-GGNN models. All invariant models succeed when the third power is taken, as expected since a connected graph's $d + 1$ power (here $d = 2$) is generically globally rigid.

Table 1: E-GGNN models like EGENNET can separate the two pairs of connected graph in Figure 2. I-GGNN models cannot separate unless power graphs are considered.

| | | Power graph | | |
|---|---|---|---|---|
| | **GNN Layer** | 1 | 2 | 3 |
| Pair a | Maximal I-GGNN | 50% | 100% | 100% |
| | SchNet Schütt et al. (2018) | $50.0 \pm 0.0$ | $100.0 \pm 0.0$ | $100.0 \pm 0.0$ |
| | DimeNet Gasteiger et al. (2020b) | $100.0 \pm 0.0$ | $100.0 \pm 0.0$ | $100.0 \pm 0.0$ |
| | SphereNet Liu et al. (2021) | $100.0 \pm 0.0$ | $100.0 \pm 0.0$ | $100.0 \pm 0.0$ |
| | SchNet $_{\text{full graph}}$ [§] | $100.0 \pm 0.0$ | $100.0 \pm 0.0$ | $100.0 \pm 0.0$ |
| | SchNet$_{\text{global feat}}$ [§] | $100.0 \pm 0.0$ | $100.0 \pm 0.0$ | $100.0 \pm 0.0$ |
| | EGENNET(us) | $100.0 \pm 0.0$ | $100.0 \pm 0.0$ | $100.0 \pm 0.0$ |
| Pair b | Maximal I-GGNN | 50% | 50% | 100% |
| | SchNet Schütt et al. (2018) | $50.0 \pm 0.0$ | $50.0 \pm 0.0$ | $100.0 \pm 0.0$ |
| | DimeNet Gasteiger et al. (2020b) | $50.0 \pm 0.0$ | $100.0 \pm 0.0$ | $100.0 \pm 0.0$ |
| | SphereNet Liu et al. (2021) | $50.0 \pm 0.0$ | $100.0 \pm 0.0$ | $100.0 \pm 0.0$ |
| | SchNet $_{\text{full graph}}$ [§] | $100.0 \pm 0.0$ | $100.0 \pm 0.0$ | $100.0 \pm 0.0$ |
| | SchNet$_{\text{global feat}}$ [§] | $100.0 \pm 0.0$ | $100.0 \pm 0.0$ | $100.0 \pm 0.0$ |
| | EGENNET (us) | $100.0 \pm 0.0$ | $100.0 \pm 0.0$ | $100.0 \pm 0.0$ |

### Unpacking the Bottleneck: Layer Requirements for Complex Graph Separation

**k-chains** Next, we consider the $k$ chain pairs suggested in Joshi et al. (2023) and illustrated in the left of Figure 2. By design, this is a pair of graphs with $k$ nodes, which can only be separated by $k - 1$ hop distances. In Joshi et al. (2023), it was shown that a depth of at least $\lfloor k/2 \rfloor + 1$ is necessary to achieve separation by E-GGNN in this setting. Note that as most nodes in the chain graphs are on the same line, this point cloud is not generic, so the fact that for large $k$, more than $d + 1$ iterations are needed does not contradict our theory. In our experiments, we include two variations of SchNet (Schütt et al., 2018): full graph and global feature. Those variations use all pairwise distances, thus showing better results than other invariant models. In the first experiment, we set $k = 4$. Theoretically, 3 blocks are needed for propagating the information, and 2 blocks are insufficient. As shown in Appendix 3, we succeed with a probability of 1 over 10 trials with 3 blocks. For checking long-term dependencies, we choose $k = 12$ again; 7 blocks are needed, and 6 is insufficient. We examined all models with $7, 8, 9, 10, 11$ blocks, each 10 times. As shown in 4, we are the only ones succeeding with 7 blocks, and all other models need a much higher number of blocks. This may suggest our model is less influenced by over-squashing Alon & Yahav (2020). In the appendix, Table 7, we add a time comparison table for all models using the 128 dimensional feature.

**Hard examples** In 5 in the appendix, we show that our model can also perfectly separate challenging examples from (Pozdnyakov et al., 2020; Pozdnyakov & Ceriotti, 2022), using the experiment setup taken from Hordan et al. (2023). These geometric graph pairs cannot be separated by I-GGNN, even on full graphs, and are also difficult to separate using equivariant methods like EGNN.

**Chemical property experiments** Finally, we check our model's ability to learn regression-invariant real-world chemical properties, a critical task in computational chemistry and material science. Learning chemical properties accurately is essential for predicting the behaviors of compounds, enabling advancements in drug discovery, material design, and environmental science. By effectively capturing these properties, the model demonstrates its potential to assist in solving complex chemical problems, where precise property prediction can reduce experimental costs and guide targeted research. As in (Zhu et al., 2024), we compare to the following models: Random Forest (Breiman, 2001), LSTM (Hochreiter & Schmidhuber, 1997), Transformer (Vaswani et al., 2017), GIN (Xu et al.,

---

[§] Here, SchNet global and full are invariant models utilizing the full graph structure (unlike other models utilizing the given combinatorial structure).

Table 2: Performance of 1D, 2D, and 3D baseline MRL models and the best results from ensemble learning strategies on 3D GNNs. The metric used is the Mean Absolute Error (MAE, ↓). The **bold** indicates the best-performing model, while underlined denotes the second-best.

| Category | Model | Drugs | | | Kraken | | | | BDE |
| --- | --- | --- | --- | --- | --- | --- | --- | --- | --- |
| | | IP | EA | $\chi$ | $B_5$ | L | $BurB_5$ | BurL | BE |
| 1D | Random Forest | 0.4987 | 0.4747 | 0.2732 | 0.4760 | 0.4303 | 0.2758 | 0.1521 | 3.0335 |
| | LSTM | 0.4788 | 0.4648 | 0.2505 | 0.4879 | 0.5142 | 0.2813 | 0.1924 | 2.8279 |
| | Transformer | 0.6617 | 0.5850 | 0.4073 | 0.9611 | 0.8389 | 0.4929 | 0.2781 | 10.0771 |
| 2D | GIN | 0.4354 | 0.4169 | 0.2260 | 0.3128 | 0.4003 | 0.1719 | 0.1200 | 2.6368 |
| | GIN+VN | 0.4361 | 0.4169 | 0.2267 | 0.3567 | 0.4344 | 0.2422 | 0.1741 | 2.7417 |
| | ChemProp | 0.4595 | 0.4417 | 0.2441 | 0.4850 | 0.5452 | 0.3002 | 0.1948 | 2.6616 |
| | GraphGPS | 0.4351 | 0.4085 | 0.2212 | 0.3450 | 0.4363 | 0.2066 | 0.1500 | 2.4827 |
| 3D | SchNet | 0.4394 | 0.4207 | 0.2243 | 0.3293 | 0.5458 | 0.2295 | 0.1861 | 2.5488 |
| | DimeNet++ | 0.4441 | 0.4233 | 0.2436 | 0.3510 | 0.4174 | 0.2097 | 0.1526 | 1.4503 |
| | GemNet | 0.4069 | 0.3922 | **0.1970** | 0.2789 | 0.3754 | 0.1782 | 0.1635 | 1.6530 |
| | PaiNN | 0.4505 | 0.4495 | 0.2324 | 0.3443 | 0.4471 | 0.2395 | 0.1673 | 2.1261 |
| | ClofNet | 0.4393 | 0.4251 | 0.2378 | 0.4873 | 0.6417 | 0.2884 | 0.2529 | 2.6057 |
| | LEFTNet | 0.4174 | 0.3964 | 0.2083 | 0.3072 | 0.4493 | 0.2176 | 0.1486 | 1.5328 |
| | Marcel | 0.4066 | 0.3910 | 0.2027 | 0.2225 | 0.3386 | 0.1589 | 0.0947 | 1.4741 |
| | EGENNET(Ours) | **0.3965** | **0.3686** | 0.2068 | **0.1878** | **0.1738** | **0.1530** | **0.0626** | **0.3170** |

2018), GIN+VN (Hu et al., 2020), ChemProp (Heid et al., 2023), GraphGPS (Rampášek et al., 2022), SchNet (Schütt et al., 2018), DimeNet++ (Gasteiger et al., 2020a), GemNet (Gasteiger et al., 2021), PaiNN (Schütt et al., 2021), ClofNet (Du et al., 2022), LEFTNet (Du et al., 2024), and Marcel (Zhu et al., 2024). Our task is learning chemical property prediction on three datasets: Drugs (Axelrod & Gomez-Bombarelli, 2022), Kraken (Gensch et al., 2022), and BDE (Meyer et al., 2018). Drugs are the largest and most challenging dataset, containing 560K samples; BDE is a medium-sized dataset with 80K datasets; and Kraken is the smallest, with 22K samples. The baseline model performance, data split, and training procedure are taken from Zhu et al. (2024). In particular, we ran each experiment three times and reported the model's test error with the best validation performance. In Table 6 in the appendix, we report the mean and standard deviation.

As we see in Table 2, we show comparable or improved results in all tasks. For example, in BDE, we have an improvement of $78\%$, in Kraken property *burL* $33\%$, and in the property $L$ $48\%$. Remarkably, our method outperforms competing more complex methods like GemNet, which employs high-order irreducible representations. This can be explained by the generic completeness of our method, coupled with results we will show in the appendix Table 8, which shows that these datasets have a small number of norm-repetitions, indicating that they are 'close to generic' .

## 7 LIMITATION AND FUTURE WORK

In this work, we characterized the generic expressive power of I-GGNN and E-GGNN, showed that EGENNET is a generically maximally expressive E-GGNN, and showed the effectiveness of this architecture on chemical regression tasks. A limitation of our work is that it may struggle with symmetric and non-generic input. Future work could consider assessing the expressive power of non-MPNN models like geometric 2-WL on sparse graphs to address such input efficiently.

## 8 ACKNOWLEDGMENTS

We thank Snir Hordan for reviewing our manuscript and for the helpful discussions and suggestions. We would like to thank Idan Tankel for his technical help and support. N.D. and Y.S. are funded by Israeli Science Foundation grant no. 272/23.

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

## A    RELATED WORK

This section discusses related work that is not discussed in the main text.

**Equivariant models**    In the context of learning on geometric graphs, models should be equivariant to the joint action of permutations, rotations, and translations. Besides the models discussed in the paper, some prominent models with this equivariant structure include Cormorant (Anderson et al., 2019), Vector Neurons (Deng et al., 2021), and SE(3) transformers (Fuchs et al., 2020).

**Completeness, universality and generic completeness**    The main text discusses related works on completeness focused on methods coming from the geometric learning community. Other complete methods, generally using similar ideas, appeared independently in the more chemistry-oriented community(Kurlin, 2023; Widdowson & Kurlin, 2023; Shapeev, 2016; Nigam et al., 2020). Completeness in $2D$ was achieved in Bökman et al. (2022), and completeness with respect to rigid motions (without permutations) was achieved by Comenet (Wang et al., 2022). Comenet also uses some genericity assumptions (e.g., that there is a well-defined notion of nearest neighbors).

The main text discussed how generic completeness can be achieved using I-GGNN when the graph is globally rigid. A similar result can be obtained when considering the list of all distances (Garamvölgyi et al., 2022). Generic completeness for full graphs can also be obtained using more complex distance-based features (Widdowson et al., 2022) or shape principal axes (Puny et al., 2021) (under the generic assumption that the principal eigenvalues are distinct).

**Comparison with Wang & Zhang (2022)**    Generic completeness without the full graph assumption, employing only the assumption that local neighborhoods are not symmetric, is discussed by Wang & Zhang (2022). However, there are notable differences in key aspects of the results. Specifically, Theorem 4.2 in Wang & Zhang (2022) demonstrates that their proposed equivariant model can identify generic molecules using $L$ message-passing iterations, where $L$ is the graph's diameter. In contrast, our Theorem 4.1 requires only $d+1$ iterations (where $d = 3$ in typical applications, resulting in a total of just four iterations).

Moreover, while Theorem 4.2 in Wang & Zhang (2022) focuses on recovering the 3D structure, it does not address the combinatorial structure. Our Theorem 4.1, however, guarantees the recovery of both combinatorial and geometric structures in the generic case.

Additionally, our Theorem 3.1 precisely describes the conditions under which invariant models are, and are not, generically complete—a result not covered in Wang & Zhang (2022).

## B    IMPLEMENTATION DETAILS

Here, we detail the hyper-parameter choice. For all experiments, we use AdamW optimizer Loshchilov & Hutter (2017). We use a varying weight decay of $1e^{-4}, 1e^{-5}$ and learning rate of $1e^{-4}$. Reduce On Platue scheduler is used with a rate decrease of 0.75 with the patience of 15 epochs. We use two blocks with 40 channels in the challenging point cloud separation experiments. For *k-chain* experiments, we repeat each experiment 10 times, with 10 different seeds, and set the number of channels to be 60, and a varying number of blocks, as detailed in the table. For power graph experiments, we repeat each experiment 10 times, with 10 different seeds, and set the number of channels to be 60 and 3 blocks. For chemical property experiments, all the procedure is taken from Zhu et al. (2024): each dataset is partitioned randomly into three subsets: 70% for training, 10% for validation, and 20% for test. We set the number of channels to be 256 and used 6 blocks. Each model is trained over $1,500$ epochs. Experiments are repeated three times for all eight regression targets, and the results reported correspond to the model that performs best on the validation set in terms of Mean Absolute Error (MAE). In contrast to the experiments in Zhu et al. (2024), we didn't consider the EE dataset as it's currently not public on GitHub. We utilize PyTorch and PyTorch-Geometric to implement all deep learning models. All experiments are conducted on servers with NVIDIA A40 GPUs, each with 48GB of memory. All dataset details, including statistics, can be found in Zhu et al. (2024).

## C  BUILDING PERFECT I-GGNN

Here, we describe in detail how to construct a perfect I-GGNN. We will assume that we consider geometric graphs with up to $N$ nodes. The main ingredients needed are aggregation functions and READOUT function that are injective on the space of all multisets of size $\leq N$. Amir et al. (2023) proved that the sum of projections and analytic activation can be used to construct such layers. Formally, be $\mathbb{M}$ semi-analytic set of dimension $D$, and $N$ vectors in $\mathbb{R}^d$ denoted by $\boldsymbol{X}$. Be $A \in \mathbb{R}^{(2D+1)\times d}, b \in \mathbb{R}^{2D+1}$, $\sigma : \mathbb{R} \to \mathbb{R}$ analytic non-polynomial activation, then for Lebesgue almost every $A, b$

$$f_{A,b}(\boldsymbol{X}) = \sum_{i=1}^{N} \sigma(A \cdot \boldsymbol{X}_i + b)$$

is injective up to permutation. Thus, the composition of such blocks and such READOUT function with dimension $D = d \cdot N$ yields a maximally expressive model on all input domain. Here, $D$ is the intrinsic dimension of all features, ultimately produced by the $D$ dimensional input space.

## D  SEPARATION POWER OF FULLY MAXIMALLY EXPRESSIVE E-GGNN

In the main text, we proposed EGENNET a simple *generically* maximal expressive E-GGNN. In this section, we show that if we have a maximally expressive E-GGNN on all input domain, we can reconstruct a full graph up to group action, and if the graph is connected, we can reconstruct its point cloud. This indicates that fully maximally expressive E-GGNN is as challenging as achieving completeness, which generally requires complex methods like 2-WL-based GNNs.

**Theorem D.1.** *Let $\mathcal{G} = (\boldsymbol{A}, \boldsymbol{X})$ be a geometric graph, where $\boldsymbol{A}$ is the full graph on $n$ nodes, and $\boldsymbol{X}$ is an arbitrary $d \times n$ matrix, then one iteration of maximally expressive E-GGNN can reconstruct $\boldsymbol{X}$.*

*Proof.* Assume we run one iteration of E-GGNN on our geometric graph, then for each node $i$, we know up to some rotation $Q_i$

$$\{\!\!\{ Q_i \cdot (x_i - x_j) | j \in [n] \}\!\!\}$$

Choosing $i = 1$, and $x_i = 0$, we have all other positions $Q_i \cdot x_j, j \neq i$, thus up to translation (as we set $x_i$), rotation $Q_i$ and node permutation, we can reconstruct our point cloud. □

We now show that using $n$ iterations of E-GGNN can reconstruct the point cloud of each connected graph.

**Theorem D.2.** *Assuming a connected geometric graph $\mathcal{G} = (\boldsymbol{A}, \boldsymbol{X})$ with $n$ nodes, then $n$ iteration of maximally expressive E-GGNN can reconstruct the point cloud.*

*Proof.* Assume we run $n$ iterations of a maximally expressive E-GGNN on our connected graph. Then, by Theorem 4.1, we can reconstruct for each node $i$ its $n$-hop up to rotation $Q_i$. As the $n$-hop of a node is all other nodes (the graph's diameter is at most $n$), we reconstruct the full geometric graph up to rotation, and by theorem D.1, we can reconstruct the point cloud. □

## E  I-GGNN+

Here, we present the definition of I-GGNN presented by Joshi et al. (2023), denoted by I-GGNN+, and show that it can separate all geometric full graphs. After, we explain why this model is strictly stronger than our I-GGNN.

Joshi et al. (2023) defines (neglecting the update step also used there)

$$\boldsymbol{v}_i^{(t+1)} = f^t \left( \{\!\!\{ \boldsymbol{v}_j^{(t)}, \boldsymbol{x}_i - \boldsymbol{x}_j, | j \in \mathcal{N}_i \}\!\!\} \right). \tag{7}$$

Such that $f^t$ holds that for all $Q \in O(d)$

$$f^t \left( \{\!\!\{ \boldsymbol{v}_j^{(t)}, \boldsymbol{x}_i - \boldsymbol{x}_j, | j \in \mathcal{N}_i \}\!\!\} \right) = f^t \left( \{\!\!\{ \boldsymbol{v}_j^{(t)}, Q \cdot (\boldsymbol{x}_i - \boldsymbol{x}_j), | j \in \mathcal{N}_i \}\!\!\} \right)$$

I-GGNN+ is maximally expressive if $f^t$ holds:

$$f^t\left(\{\!\{\boldsymbol{v}_j^{(t)}, \boldsymbol{x}_i - \boldsymbol{x}_j, | \, j \in \mathcal{N}_i\}\!\}\right) = f^t\left(\{\!\{\boldsymbol{v}_j^{(t)}, \hat{\boldsymbol{x}}_i - \hat{\boldsymbol{x}}_j, | \, j \in \mathcal{N}_i\}\!\}\right) \iff$$

$$\exists Q \in O(3) : \{\!\{\boldsymbol{v}_j^{(t)}, \boldsymbol{x}_i - \boldsymbol{x}_j, | \, j \in \mathcal{N}_i\}\!\} = \{\!\{\boldsymbol{v}_j^{(t)}, Q \cdot (\hat{\boldsymbol{x}}_i - \hat{\boldsymbol{x}}_j), | \, j \in \mathcal{N}_i\}\!\}$$

We now prove that if we run one iteration of maximally expressive I-GGNN+ on a full graph, we can reconstruct it.

**Theorem E.1.** *Let $\mathcal{G} = (\boldsymbol{A}, \boldsymbol{X})$ be a geometric graph, where $\boldsymbol{A}$ is the full graph on $n$ nodes, and $\boldsymbol{X}$ is an arbitrary $d \times n$ matrix, then one iteration of maximally expressive I-GGNN+ can reconstruct $\boldsymbol{X}$.*

*Proof.* Assume we run one iteration of I-GGNN+. Then, for each node $i$, we have

$$f^t\left(\{\!\{\boldsymbol{v}_j^{(t)}, \boldsymbol{x}_i - \boldsymbol{x}_j, | \, j \in [n]\}\!\}\right)$$

then up to some rotation $Q_i \in O(3)$, we know

$$\{\!\{Q_i \cdot (\boldsymbol{x}_i - \boldsymbol{x}_j), | \, j \in [n]\}\!\}$$

Then, we can reconstruct the point cloud by Theorem D.1. $\qquad\square$

It's easy to see that I-GGNN+ can simulate our I-GGNN as the norm function is invariant under multiplication by $Q \in O(3)$. But (Pozdnyakov et al., 2020; Pozdnyakov & Ceriotti, 2022) gave examples of full-graphs each I-GGNN can't separate, but by what we proved above, maximally expressive I-GGNN+ can separate them. Thus, I-GGNN+ is strictly more powerful than I-GGNN in its expressive power.

# F  APPROXIMATION AND SEPARATION

It is known that a tight connection exists between the approximation of invariant functions and complete invariant models Hordan et al. (2023).

In the context of our paper, we discussed generic completeness. In particular, we showed in Corollary 5.2 that EGENNET with random parameters is complete on $\mathbb{G}_{\text{generic}}(d, N)$, which denotes the set of all geometric graphs $\mathcal{G} = (\boldsymbol{A}, \boldsymbol{X})$ with exactly $N$ nodes, $\boldsymbol{A}$ connected, and $\boldsymbol{X} \in \mathbb{R}^{d \times n}$ generic. Accordingly, we can obtain the following theorem:

**Theorem F.1.** *Let $K \subseteq \mathbb{G}_{\text{generic}}(d, N)$ be a compact set, $f^{sep} : K \to R^m$ a continuous function invariant to rotations, translations, and permutations. Then $f^{sep}$ is injective on $K$ (up to symmetries) if and only if every continuous invariant $f : K \to R^M$ can be approximated uniformly on $K$ using functions of the form $\mathcal{N}(f^{sep})$, where $\mathcal{N}$ is a fully connected neural network.*

*Proof.* Assume, on the one hand, $f^{sep}$ is injective on $K$ (up to symmetries). Given a continuous $f : K \to R^M$ we want to approximate, then, by Hordan et al. (2023), $\forall \epsilon > 0 \, \exists$ neural network $\mathcal{N}$:

$$sup_{\mathcal{G} \in K} |f(\mathcal{G}) - \mathcal{N}(f^{sep}(\mathcal{G}))| < \epsilon$$

On the other hand, assume by contradiction there exists $\mathcal{G}_1 = (\boldsymbol{A}_1, \boldsymbol{X}_1)$ and $\mathcal{G}_2 = (\boldsymbol{A}_2, \boldsymbol{X}_2)$ which are not related by translation, rotation or permutation, such that $f^{sep}(\mathcal{G}_1) = f^{sep}(\mathcal{G}_2)$.

Define $f(\mathcal{G}) = Inf_{g \in G} \|(\boldsymbol{A}, \boldsymbol{X}) - g \cdot (\boldsymbol{A}_1, \boldsymbol{X}_1)\|$, where $\mathcal{G} = (\boldsymbol{A}, \boldsymbol{X})$, and $G$ is the group of permutations, rotations, and translation. Note that this minimum is obtained as the product group of rotation and permutation is compact, and the translation can be bounded by the sum of the norms of the two graphs. Then, according to the Tikhonov theorem, our group is compact, and we take a minimum continuous function over a compact set. Note that $f$ is an invariant, continuous function such that $f(\mathcal{G}_1) = 0$, and $f(\mathcal{G}_2) > 0$, but for each neural network $\mathcal{N}$, $\mathcal{N}(f^{sep}(\mathcal{G}_1)) = \mathcal{N}(f^{sep}(\mathcal{G}_1))$, and we can't approximate our desired function $f$ with $\epsilon = \frac{f(\mathcal{G}_2)}{2}$ accuracy, yielding a contradiction. $\qquad\square$

## G PROOFS

In this section, we restate and prove theorems that have not been fully proved in the main text.

**Theorem 3.1.** *[expressive power of I-GGNN] Let $d$ be a natural number. Let $F$ be an I-GGNN. Let $\boldsymbol{A}$ be a graph not generically globally rigid on $\mathbb{R}^d$. Then, $F$ generically fails to identify $\boldsymbol{A}$.*
*Conversely, if $\boldsymbol{A}$ is generically globally rigid on $\mathbb{R}^d$ and $F$ is a maximally expressive I-GGNN with depth $T = 1$, then $F$ generically identifies $\boldsymbol{A}$.*

*Proof.* Let $\mathcal{G} = (\boldsymbol{A}, \boldsymbol{X})$ be a geometric graph, and assume that $\boldsymbol{X}$ is generic. First, we assume that $\boldsymbol{A}$ is generically globally rigid. Assume that there is some geometric graph $\mathcal{G}'$ which is assigned the same global feature as $\mathcal{G}$ after a single iteration. We need to show the two graphs are geometrically isomorphic. The multisets of $s_i$ and $s_i'$ features in equation 2 are equal, and in particular, both graphs have the same number of nodes $n$, and by relabeling if necessary, we can assume that $s_i = s_i'$ for all $i = 1, \ldots, n$. This equation, in turn, implies that for every node $i$,

$$\{\!\{\|\boldsymbol{x}_{ij}\|, j \in \mathcal{N}_i\}\!\} = \{\!\{\|\boldsymbol{x}_{ij}'\|, j \in \mathcal{N}_i'\}\!\}$$

In particular, the $i$'th node in $\mathcal{G}$ and $\mathcal{G}'$ have the same degree, so both graphs have the same number of edges. For any fixed $(i, j)$ which is an edge of $\mathcal{G}$, the distance $\|x_{ij}\|'$ will appear exactly once in the multiset corresponding to $s_i'$ and $s_j'$ and will not appear in the other multisets. This observation implies that $(i, j)$ is also an edge in the graph $\mathcal{G}'$. Since the number of edges in both graphs is the same, we deduce that $\boldsymbol{A} = \boldsymbol{A}'$. By global rigidity, we deduce that all pairwise distances are the same, and by Satorras et al. (2021), the two graphs are geometrically isomorphic.

We mention a theorem proved in Gortler et al. (2010) for the other direction. Gortler et al. (2010) proved $\boldsymbol{A}$ is globally rigid if exists generic $\boldsymbol{X}$ such that $(\boldsymbol{A}, \boldsymbol{X})$ is globally rigid. Thus if $\boldsymbol{A}$ is not globally rigid, for *every* generic $\boldsymbol{X}$, there exists $\boldsymbol{X}'$ for whom distances along edges are preserved, but $\boldsymbol{X}$ and $\boldsymbol{X}'$ aren't geometrically isomorphic. By immediate induction on the number of layers, it can be shown that any I-GGNN cannot separate any pair of geometric graphs whose distances across edges are preserved. This concludes the proof. $\square$

**Theorem 4.1.** *[expressive power of E-GGNN] Let $d$ be a natural number. Let $F$ be an E-GGNN. Let $\boldsymbol{A}$ be a disconnected graph. Then, $F$ generically fails to identify $\boldsymbol{A}$.*
*Conversely, if $\boldsymbol{A}$ is connected and $F$ is a maximally expressive E-GGNN with depth $T \geq d + 1$, then $F$ generically identifies $\boldsymbol{A}$.*

*Proof.* Assume $\boldsymbol{A}$ is not connected. Let $\boldsymbol{X} \in \mathbb{R}^{n \times d}$. Let $\boldsymbol{V}_1, \boldsymbol{V}_2$ be two connected components of the graph, and let $\mathcal{G}'$ be the geometric graph whose adjacency matrix is $\boldsymbol{A}' = \boldsymbol{A}$, and where $\boldsymbol{X}'$ is obtained from $\boldsymbol{X}$ by applying a non-trivial rotation $Q$ not in the symmetry group of $\boldsymbol{V}_2 : \hat{\boldsymbol{x}}_i = Q \cdot \boldsymbol{x}_i$ for all $i \in V_1$, and setting $\hat{\boldsymbol{x}}_j = \boldsymbol{x}_j$ for all $j \in V_2$. The two graphs are not geometrically isomorphic by construction. However, for any E-GGNN, we can easily see recursively that all features $\boldsymbol{v}_i^{(t)}$ constructed from $\boldsymbol{X}$ and $\boldsymbol{X}'$, respectively, will be the same, and therefore, the E-GGNN will assign the same global features to $\mathcal{G}$ and $\mathcal{G}'$.

Now assume $\boldsymbol{A}$ is connected, and we are given two geometric graphs $\mathcal{G}$ and $\hat{\mathcal{G}}$ in $(\mathbb{G}_{\text{generic}}(d, N), \mathbb{G}_N(d))$. By assumption, $\boldsymbol{X}$ is generic. Assume that the E-GGNN assigns the same value of $\mathcal{G}$ and $\hat{\mathcal{G}}$. We need to show that the two graphs are geometrically isomorphic. First, we note that since the two graphs have the same global features, they have the same number of nodes $n$, and by applying relabeling if necessary, we can assume that

$$s_i = \hat{s}_i, \quad \forall i = 1, \ldots, n$$

which implies that for every node $i$ there is some rotation $R_i \in O(d)$ such that $\boldsymbol{v}_i^{(T)} = R_i \hat{\boldsymbol{v}}_i^{(T)}$. By rotation equivariance, we know that $R_i \hat{\boldsymbol{v}}_i^{(T)}$ is the feature that we would have obtained by applying the maximally expressive E-GGNN to $R_i \cdot \hat{\boldsymbol{X}}$. We deduce that

$$\{\!\{\boldsymbol{v}_j^{(T-1)}, \boldsymbol{x}_i - \boldsymbol{x}_j | j \in \mathcal{N}_i\}\!\} = \{\!\{R_i \hat{\boldsymbol{v}}_j^{(T-1)}, R_i(\hat{\boldsymbol{x}}_i - \hat{\boldsymbol{x}}_j) | j \in \hat{\mathcal{N}}_i\}\!\}$$

The equality of multisets above implies that the node $i$ has the same degree in $\boldsymbol{A}$ and $\hat{\boldsymbol{A}}$. Since this is true for all nodes, $\boldsymbol{A}$ and $\hat{\boldsymbol{A}}$ have the same number of edges. Moreover

$$\{\!\{\|\boldsymbol{x}_i - \boldsymbol{x}_j\|, j \in \mathcal{N}_i\}\!\} = \{\!\{\|\boldsymbol{x}_i - \boldsymbol{x}_j\|, j \in \hat{\mathcal{N}}_i\}\!\}$$

Since $\boldsymbol{X}$ is generic, for every edge $(i,j) \in \boldsymbol{A}$ the distance $\|\boldsymbol{x}_i - \boldsymbol{x}_j\|$ will appear only in the 1-hop multi-sets corresponding to the $i$ and $j$ nodes. Thus, they will appear only once in the 1-hop multisets corresponding to the second graph's $i$ and $j$ nodes. This implies that $(i,j)$ is an edge in $\hat{\boldsymbol{A}}$ too, and

$$R_i(\hat{\boldsymbol{x}}_i - \hat{\boldsymbol{x}}_j) = \boldsymbol{x}_i - \boldsymbol{x}_j, \forall j \in \mathcal{N}_i.$$

Since the two graphs have the same number of edges, it follows that $\boldsymbol{A} = \hat{\boldsymbol{A}}$. We now show recursively that for all $1 \leq t \leq T - 1$, we have $\boldsymbol{v}_j^{T-t} = R_i \hat{\boldsymbol{v}}_j^{T-t}$ and $\boldsymbol{x}_i - \boldsymbol{x}_j = R_i(\hat{\boldsymbol{x}}_i - \hat{\boldsymbol{x}}_j)$ for all $j$ that is a $t$-hop neighbor of $i$ (we say $j$ is an $t$-hop neighbor of $i$, if there is a path of length at most $t$ from $i$ to $j$).

We have already proved the claim for $t = 1$. Now we assume correctness for $t$ and show for $t + 1$. Since by the induction hypothesis we know that $\boldsymbol{v}_j^{T-t} = R_i \hat{\boldsymbol{v}}_j^{T-t}$ for every $j$ which is $t$ hop neighbor of $i$, we deduce the multi-set equality

$$\{\!\{\boldsymbol{v}_k^{T-(t+1)}, \boldsymbol{x}_j - \boldsymbol{x}_k | k \in \mathcal{N}_j\}\!\} = \{\!\{R_i \hat{\boldsymbol{v}}_k^{T-(t+1)}, R_i(\hat{\boldsymbol{x}}_j - \hat{\boldsymbol{x}}_k) | k \in \mathcal{N}_j\}\!\}$$

Since we know that the pairwise norms of $\boldsymbol{X}$ are all distinct to to the genericity of $\boldsymbol{X}$, and also that for every edge $(j,k)$ we have the norm equality $\|\boldsymbol{x}_j - \boldsymbol{x}_k\| = \|\hat{\boldsymbol{x}}_j - \hat{\boldsymbol{x}}_k\|$, we see that necessarily

$$\boldsymbol{v}_k^{T-(t+1)} = R_i \hat{\boldsymbol{v}}_k^{T-t+1}, \text{and } \boldsymbol{x}_j - \boldsymbol{x}_k = R_i(\hat{\boldsymbol{x}}_j - \hat{\boldsymbol{x}}_k).$$

Now, every $k$, an $t + 1$ hop neighbor of $i$, is connected by an edge to a node $j$, an $t$ hop neighbor of $i$. By the induction assumption we know that $\boldsymbol{x}_i - \boldsymbol{x}_j = R_i(\hat{\boldsymbol{x}}_i - \hat{\boldsymbol{x}}_j)$. We deduce that

$$\boldsymbol{x}_i - \boldsymbol{x}_k = \boldsymbol{x}_i - \boldsymbol{x}_j + \boldsymbol{x}_j - \boldsymbol{x}_k = R_i(\hat{\boldsymbol{x}}_i - \hat{\boldsymbol{x}}_j + \hat{\boldsymbol{x}}_j - \hat{\boldsymbol{x}}_k) = R_i(\hat{\boldsymbol{x}}_i - \hat{\boldsymbol{x}}_k)$$

for all $t + 1$ hop neighbors of $i$. Continuing with this argument inductively, we see that for all $T = d + 1$ hop neighbors $k$ of $i$, we obtain

$$\boldsymbol{x}_i - \boldsymbol{x}_k = R_i(\hat{\boldsymbol{x}}_i - \hat{\boldsymbol{x}}_k)$$

and in particular

$$\|\boldsymbol{x}_i - \boldsymbol{x}_k\| = \|\hat{\boldsymbol{x}}_i - \hat{\boldsymbol{x}}_k\|$$

for all $d + 1$ hop neighbors. By global generic rigidity of the $d + 1$ power graph of a connected graph (Jordán & Tanigawa, 2022; Cheung & Whiteley, 2005), it follows that the two graphs are geometrically isomorphic. Thus, as previously, we reconstructed the combinatorial graph, and we now reconstructed the geometric graphs; we are done. $\qquad\square$

### G.1 PROOF OF MAXIMAL EXPRESSIVITY OF EGENNET

This subsection is devoted to proving Theorem 5.1 on the generic maximal expressivity of EGENNET. This proof will require some background on the finite witness theorem from Amir et al. (2023):

**Background on the finite witness theorem**   Before we prove the theorem, we review some definitions and results from Amir et al. (2023), which we will use for the proof. The main tool we require from Amir et al. (2023) is Theorem A.2 from that paper, the finite witness theorem. We state it here in the special case where the parameter space is Euclidean, and the function $F$ is analytic in the parameters:

**Theorem G.1** (Finite Witness Theorem, Amir et al. (2023)). *Let $\mathbb{M} \subseteq \mathbb{R}^p$ be a $\sigma$-sub-analytic set of dimension $D$. Let $F : \mathbb{M} \times \mathbb{R}^q \to \mathbb{R}$ be a $\sigma$-sub-analytic function, and assume that $F(\boldsymbol{z}; \boldsymbol{\theta})$ is analytic as a function of $\boldsymbol{\theta}$ for all fixed $\boldsymbol{z} \in \mathbb{M}$. Define the set*

$$\mathcal{N} = \{\boldsymbol{z} \in \mathbb{M}| \quad F(\boldsymbol{z}; \boldsymbol{\theta}) = 0, \ \forall \boldsymbol{\theta} \in \mathbb{R}^q\}.$$

*Then for Lebesgue almost every* [¶]$\left(\boldsymbol{\theta}^{(1)}, \ldots, \boldsymbol{\theta}^{(D+1)}\right) \in \mathbb{R}^{q \times (D+1)}$,

$$\mathcal{N} = \{\boldsymbol{z} \in \mathbb{M}| \quad F(\boldsymbol{z}; \boldsymbol{\theta}^{(i)}) = 0, \ \forall i = 1, \ldots D + 1\}. \tag{8}$$

---

[¶]The original statement in Amir et al. (2023) discussed *generic* $\left(\boldsymbol{\theta}^{(1)}, \ldots, \boldsymbol{\theta}^{(D+1)}\right) \in \mathbb{W}^{D+1}$, in the sense defined in Amir et al. (2023). In particular, this implies the "Lebesgue almost every" statement we used here.

**Remark:** We note that this theorem also holds when the output of $F$ is a vector in $\mathbb{R}^d$, which is what we will use in our proofs below. To see the theorem holds for a function $\hat{F}: \mathbb{M} \times \mathbb{R}^q \to \mathbb{R}^d$, we can define a new parametric function $F$ with additional parameters $a \in \mathbb{R}^d$,

$$F(\boldsymbol{z}; \boldsymbol{\theta}, a) = \langle \hat{F}(\boldsymbol{z}; \boldsymbol{\theta}), a \rangle$$

and use the theorem for $F$, which has scalar output, to deduce the theorem for $\hat{F}$.

The finite witness theorem assumes the domain is a $\sigma$ sub-analytic set. The full definition of $\sigma$-sub-analytic sets is rather involved, and we refer the reader to Appendix A in Amir et al. (2023) for a full definition. Rather than giving a full definition here, we will recall the properties of these sets, which we will need.

The class of $\sigma$ subanaltyic sets in $\mathbb{R}^d$ is rather large. In particular, it contains all open sets in $\mathbb{R}^d$, all semi-algebraic sets in $\mathbb{R}^d$ (sets defined by polynomial equalities and inequalities, like spheres and linear sub-spaces), as well as sets defined by analytic equality and inequality constraints. The class of $\sigma$ subanaltyic sets is closed under countable unions and finite intersections.

$\sigma$-sub-analytic sets are always a countable union of smooth manifolds. The dimension of a $\sigma$ sub-analytic set is the maximal dimension of the manifolds composing it.

A $\sigma$-sub-analytic function is a function whose graph is a $\sigma$ sub-analytic set. Analytic functions are, in particular, $\sigma$-sub-analytic. When a $\sigma$ sub-analytic function is applied to a $\sigma$ sub-analytic set, the image is a $\sigma$ sub-analytic set whose dimension is no larger than the dimension of the $\sigma$ sub-analytic domain Amir et al. (2023).

We can now restate and prove Theorem 5.1:

**Theorem 5.1.** *Let $d, N, T$ be natural numbers. Let $F$ be a maximally expressive E-GGNN of depth $T$, and let $F_\theta$ denote the EGENNET architecture with depth $T$ and $C = 2Nd + 1$ channels. Then, for Lebesgue almost every choice of network parameters $\theta$, we have that for all generic $\mathcal{G}, \hat{\mathcal{G}} \in \mathbb{G}_N(d)$,*

$$F(\mathcal{G}) = F(\hat{\mathcal{G}}) \iff F_\theta(\mathcal{G}) = F_\theta(\hat{\mathcal{G}})$$

*Proof.* In the proof, we consider the set of all generic geometric graphs in $\mathbb{G}_N(d)$. We will prove the theorem for an even larger set: the set $\mathbb{G}_N^{\text{distinct}}(d)$ of all geometric graphs in $\mathbb{G}_N(d)$, which additionally satisfies that for any distinct edges $(i, j)$ and $(s, t)$ in the graph, $0 < \|\boldsymbol{x}_i - \boldsymbol{x}_j\| \neq \|\boldsymbol{x}_s - \boldsymbol{x}_t\|$. Since polynomial inequalities with integer coefficients define it, we note that $\mathbb{G}_N^{\text{distinct}}(d)$ contains *all generic* graphs with $\leq N$ nodes. The advantage of considering $\mathbb{G}_N^{\text{distinct}}(d)$ is that the set of all geometric graphs in $\mathbb{G}_N^{\text{distinct}}(d)$ of the same cardinality $n \leq N$ can be thought of as a subset of $\mathbb{R}^{n \times n} \oplus \mathbb{R}^{n \times d}$. Since a constant number of polynomial inequalities defines it, it is a semi-algebraic set, and hence, in particular, $\sigma$-subanaltyic, so the finite witness theorem can be applied. In contrast, it is unclear that the set of all generic points of fixed dimension is a $\sigma$ subanaltyic set.

Our goal is to show that for the networks described in the theorem, for almost every choice of parameter values, all functions described in the procedure are injective. To be more accurate, let us introduce some notation given an arbitrary pair of geometric graphs

$$\mathcal{G} = (\boldsymbol{A}, \boldsymbol{X}), \quad \hat{\mathcal{G}} = (\hat{\boldsymbol{A}}, \hat{\boldsymbol{X}})$$

in our domain $\mathbb{G}_N^{\text{distinct}}(d)$, we denote the features corresponding to the first graph by $\boldsymbol{v}_i^{(t)}, \boldsymbol{s}_i$ and $s^{\text{global}}$ as in the main text, and the features corresponding to the second graph by $\hat{\boldsymbol{v}}_i^{(t)}, \hat{\boldsymbol{s}}_i$ and $\hat{s}^{\text{global}}$. The neighborhood of a node $k$ in $\hat{\mathcal{G}}$ is denoted by $\hat{\mathcal{N}}_k$. The number of nodes in the two graphs is denoted by $n$ and $m$, respectively.

We need to show that for all $t = 0, \ldots, T - 1$ and all nodes $i$ of $\mathcal{G}$ and $k$ of $\hat{\mathcal{G}}$,

$$\boldsymbol{v}_i^{(t+1)} = \hat{\boldsymbol{v}}_k^{(t+1)} \text{ if and only if } \{\!\{(\boldsymbol{x}_i - \boldsymbol{x}_j, \boldsymbol{v}_j^{(t)}), j \in \mathcal{N}_i\}\!\} = \{\!\{(\hat{\boldsymbol{x}}_k - \hat{\boldsymbol{x}}_j, \hat{\boldsymbol{v}}_j^{(t)}) | j \in \hat{\mathcal{N}}_k\}\!\} \quad (9)$$

$$\boldsymbol{s}_i = \hat{\boldsymbol{s}}_k \text{ if and only if } \exists \boldsymbol{Q} \in O(d), \boldsymbol{Q}\boldsymbol{v}_i^{(T)} = \hat{\boldsymbol{v}}_k^{(T)} \quad (10)$$

$$s^{\text{global}} = \hat{s}^{\text{global}} \text{ if and only if } \{\!\{\boldsymbol{s}_i | i = 1, \ldots, n\}\!\} = \{\!\{\hat{\boldsymbol{s}}_i | i = 1, \ldots, m\}\!\} \quad (11)$$

Our first step for reaching this result is using the finite witness theorem, similar to what was done in Hordan et al. (2023). We note that the features we are interested in are composed of $C = 2Nd + 1$ coordinates, denoted by $q = 1, \ldots, C$, which are of the general form

$$v_{i,q}^{(t+1)} - \hat{v}_{i,q}^{(t+1)} = F_t(\boldsymbol{X}, \boldsymbol{V}^{(t)}, \hat{\boldsymbol{X}}, \hat{\boldsymbol{V}}^{(t)}; \alpha_{q,t}), \quad t = 0, \ldots, T-1 \tag{12}$$

$$s_{i,q}^{(t+1)} - \hat{s}_{i,q}^{(t+1)} = F_T(\boldsymbol{V}^{(T)}, \hat{\boldsymbol{V}}^{(t)}; \alpha_{q,T}) \tag{13}$$

$$s_q^{\text{global}} - \hat{s}_g^{\text{global}} = F_{T+1}(\boldsymbol{s}_1, \ldots, \boldsymbol{s}_n, \hat{\boldsymbol{s}}_1, \ldots, \hat{\boldsymbol{s}}_m; \alpha_{q,T+1}) \tag{14}$$

where $\alpha_{q,t}$ denote the parameters of the appropriate function, $\boldsymbol{V}^{(t)}$ is the concatenation of all $n$ features $\boldsymbol{v}_i^{(t)}$, and $\hat{\boldsymbol{V}}^{(t)}$ is the concatenation of all $m$ features $\hat{\boldsymbol{v}}_k^{(t)}$. In essence, we claim that by the finite witness theorem, for almost every choice of network parameters, we will have the following equalities for all geometric graphs $\mathcal{G}, \hat{\mathcal{G}}$ satisfying the theorem's assumptions:

$$\boldsymbol{v}_i^{(t+1)} - \hat{\boldsymbol{v}}_k^{(t+1)} = 0 \text{ if and only if } F_t(\boldsymbol{X}, \boldsymbol{V}^{(t)}, \hat{\boldsymbol{X}}, \hat{\boldsymbol{V}}^{(t)}; \alpha_t) = 0, \forall \alpha \tag{15}$$

$$\boldsymbol{s}_i - \hat{\boldsymbol{s}}_k = 0 \text{ if and only if } F_T(\boldsymbol{V}^{(T)}, \hat{\boldsymbol{V}}^{(t)}; \alpha), \forall \alpha \tag{16}$$

$$s^{\text{global}} - \hat{s}^{\text{global}} = 0 \text{ if and only if } F_{T+1}(\boldsymbol{s}_1, \ldots, \boldsymbol{s}_n, \hat{\boldsymbol{s}}_1, \ldots, \hat{\boldsymbol{s}}_m; \alpha) = 0, \quad \forall \alpha \tag{17}$$

In other words, the finite witness theorem enables us to reduce the problem to the problem of showing that some choice of network parameter exists, which enables separation.

Some explanation is needed as to the details of how the finite witness theorem is applied. It is convenient to think of the network parameters being chosen recursively. For $t = 0$, the finite witness theorem allows us to choose the parameters $\alpha_{q,0}, q = 1, \ldots, C$ in equation 12 so that the theorem applies. Formally, to do this, we need to split into a finite number of cases: once the number of nodes $n$ and $m$ in the two graphs are specified, as well as the combinatorial graphs $\boldsymbol{A}, \hat{\boldsymbol{A}}$ and the nodes $i, k$, we can think of $F_0$ as a function which is analytic in the parameters, and $\sigma$-subanalytic on its domain (the norms used in the network are semi-algebraic functions, and hence $\sigma$-subnaltyic functions. All other functions used in the network are analytic). The function $F_0$ operates on the domain of pairs $\boldsymbol{X} \in \mathbb{R}^{d \times n}, \hat{\boldsymbol{X}} \in \mathbb{R}^{d \times m}$ which satisfy a finite number of polynomial inequalities. This domain is semi-algebraic and has dimensions $d(n + m) \leq 2Nd$. Therefore, we can apply the finite witness theorem with $C = 2Nd + 1$. Next, once the parameters $\alpha_{q,t}$ are chosen for all $t < t'$, we can choose the parameters $\alpha_{q,t'}$ again using the finite witness theorem. Central to the usage here is that, once the previous parameters were fixed, the features in the $t'$ step are an image of a $\sigma$-subanaltyic function applied to a pair $(\boldsymbol{X}, \hat{\boldsymbol{X}})$. Therefore, the intrinsic dimension of the $t'$ generation features is no larger than $2Nd + 1 = C$ (for more on this argument, see Hordan et al. (2023)). This concludes the recursive definition of parameter choice. By Fubini's theorem, one can verify that, in fact, for Lebesgue almost every parameter will be 'good.' It remains to show that

> The right hand side of equation 9 holds if and only if the right hand side of equation 15 holds (18)

> The right hand side of equation 10 holds if and only if the right hand side of equation 16 holds (19)

> The right hand side of equation 11 holds if and only if the right hand side of equation 17 holds (20)

We now reformulate these three claims as three lemmas.

We use some fixed analytic non-polynomial function $\sigma$ in the following lemmas. We say that $\phi$ is a *basic analytic network* if it is of the form

$$\phi(\boldsymbol{s}; a, b) = \sigma(a \cdot \boldsymbol{s} + b)$$

for some $a \in \mathbb{R}^d, b \in \mathbb{R}$.

Our first lemma will show 18, where $z_j$ will assume the role of $\boldsymbol{x}_i - \boldsymbol{x}_j$.

**Lemma G.2.** Let $z_1, \ldots, z_n$ and $\hat{z}_1, \ldots, \hat{z}_m$ be non-zero vectors in $\mathbb{R}^d$. Additionally assume that $\|z_1\| < \|z_2\| < \ldots < \|z_n\|$ and $\|\hat{z}_1\| < \|\hat{z}_2\| < \ldots < \|\hat{z}_m\|$.

Let $\boldsymbol{v}_1, \ldots, \boldsymbol{v}_n$ and $\hat{\boldsymbol{v}}_1, \ldots, \hat{\boldsymbol{v}}_m$ be matrices in $\mathbb{R}^{d \times C}$, and denote $\boldsymbol{v}_i = (v_{i,1}, \ldots, v_{i,C})$ and $\hat{\boldsymbol{v}} = (\hat{v}_{i,1}, \ldots, \hat{v}_{i,C})$. If for all basic analytic neural networks $\phi^{(0)}, \ldots, \phi^{(C)}$ we have that

$$\sum_{j=1}^{n} \left( \phi^{(0)}(\|z_j\|)(z_j) + \sum_{c=1}^{C} \phi^{(c)}(\|z_j\|)v_{jc} \right) - \sum_{j=1}^{m} \left( \phi^{(0)}(\|\hat{z}_j\|)(\hat{z}_j) + \sum_{c=1}^{C} \phi^{(c)}(\|\hat{z}_j\|)\hat{v}_{jc} \right) = 0 \tag{21}$$

then

$$\{\!\{(z_1, \boldsymbol{v}_1), \ldots (z_n, \boldsymbol{v}_n)\}\!\} = \{\!\{(\hat{z}_1, \hat{\boldsymbol{v}}_1), \ldots (\hat{z}_m, \hat{\boldsymbol{v}}_m)\}\!\}$$

We note that this lemma is slightly stronger than we need since, in contrast with the original MPNN, we do not use the norms of $v_{i,c}$ as features.

Our second lemma will show 19

**Lemma G.3.** Let $\boldsymbol{v} = (v_1, \ldots, v_C) \in \mathbb{R}^{d \times C}$ and $\hat{\boldsymbol{v}} = (\hat{v}_1, \ldots, \hat{v}_C) \in \mathbb{R}^{d \times C}$. If for all $\theta \in \mathbb{R}^C$ we have that

$$\|\sum_{c=1}^{C} \theta_c v_c\| = \|\sum_{c=1}^{C} \theta_c \hat{v}_c\|$$

then there exists an $\boldsymbol{Q} \in O(d)$ such that $\boldsymbol{Q}v_c = \hat{v}_c$ for all $c = 1, \ldots, C$.

For proof of this lemma, see Section 3.2 in Dym & Gortler (2024).

Our third lemma will show 20

**Lemma G.4.** Let $\boldsymbol{s}_1, \ldots, \boldsymbol{s}_n$ and $\hat{\boldsymbol{s}}_1, \ldots, \hat{\boldsymbol{s}}_m$ be vectors in $\mathbb{R}^d$.

If for all basic analytic $\phi$, we have that $\sum_{i=1}^{n} \phi(\boldsymbol{s}_i) = \sum_{i=1}^{m} \phi(\hat{\boldsymbol{s}}_i)$ then $\{\!\{\boldsymbol{s}_1, \ldots, \boldsymbol{s}_n\}\!\} = \{\!\{\hat{\boldsymbol{s}}_1, \ldots, \hat{\boldsymbol{s}}_m\}\!\}$.

For proof of this lemma, see Proposition 3.2 in Amir et al. (2023).

It remains to prove Lemma G.2. We note that if equation 21 holds for all basic analytic networks $\phi^{(c)}, c = 0, \ldots, C$, This equality is also true if each $\phi^{(c)}$ is a linear combination of a finite number of shallow neural networks. Next, since we are only considering the values of $\phi^{(c)}$ on a fixed finite collection of inputs, and since linear combinations of shallow neural networks with analytic (or even continuous) non-polynomial functions can approximate any continuous function uniformly on a compact set (and in particular on a finite set) Pinkus (1999), it follows that equation G.2 holds for all *continuous* $\phi^{(c)}$.

Next, we consider the norm multisets:

$$S = \{\!\{\|z_1\|, \ldots, \|z_n\|\}\!\}, \quad \hat{S} = \|\hat{z}_1\|, \ldots, \|\hat{z}_m\|.$$

We claim that $S = \hat{S}$. We first show that $S \subseteq \hat{S}$. Fix some $s$ in $\{1, \ldots, n\}$, we need to show that $\|z_s\| \in \hat{S}$. We define a continuous function $f^s$ such that $f^s(\|z_s\|) = 1$ and $f^s(\|z_i\|) = 0 = f^s(\|\hat{z}_j\|)$ for all $i \neq s$, and for all $j = 1, \ldots, m$ such that the norms $\|\hat{z}_j\|$ and $\|z_s\|$ are not equal. We claim that there exists $j$ for which the norms *are* equal. Otherwise, we choose the continuous functions $\phi^{(c)}, c = 1, \ldots, C$ to be the zero functions, and choose $\phi^{(0)}$ to be $f^s$. This will lead to the equation $z_s = 0$ in G.2, which contradicts the assumption that $z_s \neq 0$, so we see that $S \subseteq \hat{S}$. A symmetric argument shows that $\hat{S} \subseteq S$ and so $S = \hat{S}$. We deduce that $n = m$ and $\|z_i\| = \|\hat{z}_i\|$ for all $i = 1, \ldots, n$. We can now conclude the proof: fix an arbitrary $s$ in $\{1, \ldots, n\}$. Choose $\phi^{(0)} = f^s$ and take the other $\phi^{(c)}$ to be zero. Then G.2 gives

$$z_s - \hat{z}_s = 0$$

Similarly, for an arbitrary index $q$ in $\{1, \ldots, C\}$, we can choose $\phi^{(q)} = f^s$ and $\phi^{(c)} = 0$ for all $c \neq q$ in $\{0, 1, \ldots, C\}$. In this case G.2 gives

$$v_{jq} - \hat{v}_{jq} = 0.$$

This concludes the proof of the lemma and the proof of the theorem. $\qquad\square$

## G.2 COMPLEXITY

Running EGENNET for $T$ iterations with $C$ channels requires $O(C \cdot d \cdot deg_i)$ elementary arithmetic operation for every node $i$, at every iteration $t$. Overall, this requires $O(C \cdot d \cdot N \cdot d_{avg} \cdot T)$ to run EGENNET, where $d_{avg}$ denotes the average degree in the graph. This is linear in $N$, which dominates the other terms in the complexity bound. If we set $C = 2Nd + 1$ as required for maximal expressivity in Theorem 5.1, then the complexity will be quadratic in $N$, which is less by a factor of $n$ then (Delle Rose et al., 2023; Li et al., 2024a), and by a factor of $n^2$ than Hordan et al. (2023). We also note that the cardinality in the finite witness theorem, the main tool for our proof, depends on the intrinsic dimension of the input and not the ambient dimension. Thus, under the standard 'manifold assumption' Cayton et al. (2008), which asserts that data in learning tasks typically resides in a manifold of low dimension $r$, embedded in a high dimensional Euclidean space, the proof can be adapted to show that the number of channels necessary for maximal expressivity is only $2r + 1$. Thus, under the manifold assumption, the complexity of EGENNET is linear in the number of nodes $N$.

## H ADDING NODE AND EDGE FEATURES

This section explains how we incorporate node and edge features into our architecture and theory.

A featured geometric graph is a 4-tuple $\mathcal{G} = (\boldsymbol{A}_1, \boldsymbol{X}_1, \boldsymbol{F}_1, \boldsymbol{E}_1)$ where as before $\boldsymbol{A}$ is an $n$ by $n$ adjacency matrix, and $\boldsymbol{X}$ is a $n$ by $d$ matrix denoting node positions. $\boldsymbol{F}$ is a $n$ by $d_n$ matrix denoting node features (without the action of rigid motions), and $\boldsymbol{E}$ is a $n \times n \times d_e$ tensor of edge features. We denote the feature at a node $i$ and edge $(i, j)$ by $f_i$ (a column of $\boldsymbol{F}$) and $f_{i,j}$, respectively.

We incorporate edge and node features into EGENNET by slightly modifying the aggregation step to

$$v_{i,q}^{(t+1)} = \sum_{j \in \mathcal{N}_i} \psi_n^{(t,q)}(f_i) \cdot \psi_e^{(t,q)}(f_{i,j}) \cdot (\phi^{(t,q,0)}(\|\boldsymbol{x}_i - \boldsymbol{x}_j\|, \|\boldsymbol{v}_j\|)(\boldsymbol{x}_i - \boldsymbol{x}_j) +$$

$$\sum_{c=1}^{k} \phi^{(t,q,c)}(\|\boldsymbol{x}_i - \boldsymbol{x}_j\|, \|\boldsymbol{v}_j\|)v_{jc}^{(t)})$$

In our analysis we will take $\psi_n^{(t,q)} : \mathbb{R}^{d_n} \to \mathbb{R}$ and $\psi_e^{(t,q)} : \mathbb{R}^{d_e} \to \mathbb{R}$ to be simple linear functions

$$\psi_n^{(t,q)}(f_i) = a_n^{(t,q)} \cdot f_i + b_n^{(t,q)}, \quad \psi_e^{(t,q)}(f_{i,j}) = a_e^{(t,q)} \cdot f_{i,j} + b_q^{(t,q)}$$

In practice, in our code, we use a slightly more complex function involving radial basis functions as in Li et al. (2024a).

To extend our analysis to the case of node and edge features, we first extend our definition of isomorphism of geometric graphs to include edge and node features.

**Definition H.1.** Given two featured geometric graphs

$$\mathcal{G} = (\boldsymbol{A}_1, \boldsymbol{X}_1, \boldsymbol{F}_1, \boldsymbol{E}_1), \quad \boldsymbol{H} = (\boldsymbol{A}_2, \boldsymbol{X}_2, \boldsymbol{F}_2, \boldsymbol{E}_2)$$

$\mathcal{G}$ and $\mathcal{H}$ are *featured geometrically isomorphic* if there exists a permutation matrix $\boldsymbol{P}$, a rotation matrix $\boldsymbol{Q}$, and a translation $\boldsymbol{t}$ such that:

1. $\boldsymbol{A}_2 = \boldsymbol{P} \cdot \boldsymbol{A}_1 \cdot \boldsymbol{P}^T$.

2. $\boldsymbol{X}_2 = \boldsymbol{P} \cdot \boldsymbol{X}_1 \cdot \boldsymbol{Q} + \boldsymbol{t}$

3. $\boldsymbol{F}_2 = \boldsymbol{P} \cdot \boldsymbol{F}_1$

4. $\boldsymbol{E}_2 = \boldsymbol{P} \cdot \boldsymbol{E}_1 \cdot \boldsymbol{P}^T$

where $\boldsymbol{P} \cdot \boldsymbol{E}_1 \cdot \boldsymbol{P}^T$ should be understood as operating element-wise on the $n_e$ matrices of dimension $n \times n$ which compose the tensor $\boldsymbol{E}$.

We can then prove the following theorem: a generalization of Theorem 5.2 to the featured graph case.

**Theorem H.2.** *Let $N, d, d_e, d_n$ be natural numbers. Let $F_\theta$ denote the EGENNET architecture for featured geometric graphs, with depth $T = d + 1$ and $C = 2Nd + d_e + d_n + 1$ channels. Then for Lebesgue almost every $\theta$, we have for all featured graphs $\mathcal{G}, \mathcal{G}' \in \mathbb{G}_N(d)$ which are generic and connected,*

$$F_\theta(\mathcal{G}) = F_\theta(\hat{\mathcal{G}}) \iff \mathcal{G}, \hat{\mathcal{G}} \text{ are featured geometrically isomorphic}$$

We note that the assumption that the geometric graph is generic is only on the position coordinates $\boldsymbol{X}$. The statement will hold for all node and edge features in this case.

*Proof.* We explain how to modify the proof of Theorem 5.1 to obtain this result. First, we note that the number of channels was adapted to take into account the higher dimensionality of the data now that node and edge features are included. The finite witness theorem can be applied with this adapted dimension, as explained in Theorem 5.1. Thus, we only need to show that, for given featured graphs $\hat{\mathcal{G}}, \mathcal{G}$ satisfying the conditions of the theorem, if for all $\theta$ we have that $F_\theta(\mathcal{G}) = F_\theta(\hat{\mathcal{G}})$, then $\mathcal{G}$ and $\hat{\mathcal{G}}$ are featured geometrically isomorphic.

Now, let us assume that, for all $\theta$ we have $F_\theta(\mathcal{G}) = F_\theta(\hat{\mathcal{G}})$. We can obtain all the results we had for the case where there were no additional features by choosing the parameters

$$a_n^{(t,q)} = 0, b_n^{(t,q)} = 1, a_e^{(t,q)} = 0, b_e^{(t,q)} = 1$$

so that $\psi_n^{(t,q)} = 1 = \psi_e^{(t,q)}$, and the effect of node and edge features on the architectures is 'cancelled out'. In particular, we can deduce that, after an appropriate permutation and rigid motion are applied to the data, the graphs have the same number of nodes $n$, and for all node $i$ and iteration $t$, we have

$$\boldsymbol{v}_i^{(t)} = \hat{\boldsymbol{v}}_i^{(t)}, \boldsymbol{x}_i = \hat{\boldsymbol{x}}_i$$

Now, by setting $\phi^{(t,q,c)} = 0$ for $c \geq 1$, and considering $\phi^{(t,q,0)}$ which depend only on the first coordinate, we can deduce that

$$\sum_{j \in \mathcal{N}_i} \left[ (\psi_n^{(t,q)}(f_i) \cdot \psi_e^{(t,q)}(f_{i,j}) - \psi_n^{(t,q)}(\hat{f}_i) \cdot \psi_e^{(t,q)}(\hat{f}_{i,j}) \right] \cdot \left( \phi^{(t,q,0)}(\|\boldsymbol{x}_i - \boldsymbol{x}_j\|)(\boldsymbol{x}_i - \boldsymbol{x}_j) \right) = 0$$

for all choices of parameters of the functions $\psi_n^{(t,q)}, \psi_e^{(t,q)}$ and $\phi^{(t,q,0)}$. As we showed in the proof of Theorem 5.1, this equation implies that this equation will also hold for any continuous $\phi^{(t,q,0)}$ (and any choice of parameters of the other functions) since the span of all such basic analytic functions is dense. In particular, for fixed $i, j$, we can choose $\phi^{(t,q,0)}$ to be a continuous functions with $\phi^{(t,q,0)}(\|\boldsymbol{x}_i - \boldsymbol{x}_j\|) = 1$ and $\phi^{(t,q,0)}(\|\boldsymbol{x}_i - \boldsymbol{x}_k\|) = 0$ if $k \neq j$. Inserting this into the last equality we obtain

$$\left[ \psi_n^{(t,q)}(f_i) \cdot \psi_e^{(t,q)}(f_{i,j}) - \psi_n^{(t,q)}(\hat{f}_i) \cdot \psi_e^{(t,q)}(\hat{f}_{i,j}) \right] \cdot (\boldsymbol{x}_i - \boldsymbol{x}_j) = 0$$

by genericity $\boldsymbol{x}_i - \boldsymbol{x}_j \neq 0$, and so we deduce that

$$\psi_n^{(t,q)}(f_i) \cdot \psi_e^{(t,q)}(f_{i,j}) - \psi_n^{(t,q)}(\hat{f}_i) \cdot \psi_e^{(t,q)}(\hat{f}_{i,j}) = 0$$

for every choice of function parameters. Setting $a_n^{(t,q)} = 0, b_n^{(t,q)} = 1, b_e^{(t,q)} = 0$ we obtain that

$$a_e \cdot (f_{ij} - \hat{f}_{ij}) = 0, \quad \forall a_e \in \mathbb{R}^{d_e}$$

and therefore $f_{ij} - \hat{f}_{ij} = 0$. Similarly, setting $a_e^{(t,q)} = 0, b_e^{(t,q)} = 1, b_n^{(t,q)} = 0$ we obtain that

$$a_n \cdot (f_i - \hat{f}_i) = 0, \quad \forall a_n \in \mathbb{R}^{d_n}$$

and therefore $f_i - \hat{f}_i = 0$. This concludes the proof of the theorem. $\qquad \square$

## I ADDITIONAL EMPIRICAL RESULTS

**k-chain results** Here, we show the results of the $k$-chain experiment for $k = 4$. As we can see, only our method EGENNET and GVP Jing et al. (2020) succeed using the minimal number of blocks. This experiment illustrates our model's ability to distinguish very sparse graphs. Next, we show results for $k = 12$. As can be seen, we are the only ones succeeding with the minimal number of blocks, and other models need a higher number of blocks to succeed in distinguishing.

Table 3: Separation Learning of $4$-chain paths Figure 2. We show the learned accuracy over ten trials for each model, depending on the number of blocks. We show the minimal needed number of blocks (3) is sufficient for separation, and only Jing et al. (2020) also succeeded (among equivariant models working on the sparse graph) all 10 trials using 3 blocks.

| | ($k = 4$-chains) | **Number of layers** | | | | |
|---|---|---|---|---|---|---|
| | **GNN Layer** | $\lfloor \frac{k}{2} \rfloor$ | $\lfloor \frac{k}{2} \rfloor + 1 = \mathbf{3}$ | $\lfloor \frac{k}{2} \rfloor + 2$ | $\lfloor \frac{k}{2} \rfloor + 3$ | $\lfloor \frac{k}{2} \rfloor + 4$ |
| Equiv. | GWL | 50% | 100% | 100% | 100% | 100% |
| | E-GNN Satorras et al. (2021) | 50.0 ± 0.0 | 50.0 ± 0.0 | 50.0 ± 0.0 | 50.0 ± 0.0 | 100.0 ± 0.0 |
| | GVP-GNN Jing et al. (2020) | 50.0 ± 0.0 | 100.0 ± 0.0 | 100.0 ± 0.0 | 100.0 ± 0.0 | 100.0 ± 0.0 |
| | TFN Thomas et al. (2018) | 50.0 ± 0.0 | 50.0 ± 0.0 | 50.0 ± 0.0 | 80.0 ± 24.5 | 85.0 ± 22.9 |
| | MACE Batatia et al. (2022) | 50.0 ± 0.0 | 90.0 ± 20.0 | 90.0 ± 20.0 | 95.0 ± 15.0 | 95.0 ± 15.0 |
| | EGENNET(ours) | 50.0 ± 0.0 | 100.0 ± 0.0 | 100.0 ± 0.0 | 100.0 ± 0.0 | 100.0 ± 0.0 |
| Inv. | IGWL | 50% | 50% | 50% | 50% | 50% |
| | SchNet Schütt et al. (2018) | 50.0 ± 0.0 | 50.0 ± 0.0 | 50.0 ± 0.0 | 50.0 ± 0.0 | 50.0 ± 0.0 |
| | DimeNet Gasteiger et al. (2020b) | 50.0 ± 0.0 | 50.0 ± 0.0 | 50.0 ± 0.0 | 50.0 ± 0.0 | 50.0 ± 0.0 |
| | SphereNet Liu et al. (2021) | 50.0 ± 0.0 | 50.0 ± 0.0 | 50.0 ± 0.0 | 50.0 ± 0.0 | 50.0 ± 0.0 |
| | SchNet $_{\text{full graph}}$[§] Schütt et al. (2018) | 100.0 ± 0.0 | 100.0 ± 0.0 | 100.0 ± 0.0 | 100.0 ± 0.0 | 100.0 ± 0.0 |
| | SchNet$_{\text{global feat}}$[§] Schütt et al. (2018) | 100.0 ± 0.0 | 100.0 ± 0.0 | 100.0 ± 0.0 | 100.0 ± 0.0 | 100.0 ± 0.0 |

**Hard examples** This section presents the results mentioned in Section 6.1. We use a dataset of four difficult-to-separate pairs of point clouds. Three (Hard1,Hard2,Hard3) are taken from Pozdnyakov et al. (2020). The last pair, 'Harder,' is taken from Pozdnyakov & Ceriotti (2022) and is an example of a non-isomorphic pair that I-GGNN models cannot distinguish. We show we succeeded in separating all tuples with a probability of $1$. The tested models are EGENNET, GramNet Hordan et al. (2023), GeoEGNN Hordan et al. (2023), EGNN Satorras et al. (2021), LinearEGNN, MACE Batatia et al. (2022), TFN Thomas et al. (2018), DimeNet Gasteiger et al. (2020b), GVPGNN Jing et al. (2020) and are taken from Hordan et al. (2023).

**Chemical property experiments with full statistics** the table for the chemical property prediction tasks in the main text, Table 6, was produced using the protocol from Zhu et al. (2024). In particular, each method was run with three different seeds. The seed that did best on the validation set was chosen, and its results on the test set were reported.

Table 6 presents each task's mean and standard deviation over 3 seeds. As we can see, in most tasks, the standard deviation is rather low, and the results are qualitatively similar to those obtained in Table 2. In particular, in the tasks in the Kraken and BDE datasets, we outperform the competitors, often

Table 4: Separation Learning of $12$-chain paths Figure 2. We show the learned accuracy over ten trials for each model depending on the number of blocks. As we can see, we are the only ones that succeed with the minimal number of blocks (7), and other models need many more layers to show high Performance.

| | ($k = 12$-chains) | **Number of layers** | | | | |
|---|---|---|---|---|---|---|
| | **GNN Layer** | $\lfloor \frac{k}{2} \rfloor$ | $\lfloor \frac{k}{2} \rfloor + 1$ | $\lfloor \frac{k}{2} \rfloor 2$ | $\lfloor \frac{k}{2} \rfloor + 3$ | $\lfloor \frac{k}{2} \rfloor + 4$ |
| Equiv. | Maximally expressiveness E-GGNN | 50% | 100% | 100% | 100% | 100% |
| | E-GNN Satorras et al. (2021) | 50.0 ± 0.0 | 50.0 ± 0.0 | 50.0 ± 0.0 | 50.0 ± 0.0 | 95 ± 15.0 |
| | GVP-GNN Jing et al. (2020) | 50.0 ± 0.0 | 50.0 ± 0.0 | 60.0 ± 20.0 | 50.0 ± 0.0 | 60.0 ± 20.0 |
| | TFN Thomas et al. (2018) | 50.0 ± 0.0 | 50.0 ± 0.0 | 50.0 ± 0.0 | 50.0 ± 0.0 | 50.0 ± 0.0 |
| | MACE Batatia et al. (2022) | 50.0 ± 0.0 | 50.0 ± 0.0 | 50.0 ± 0.0 | 50.0 ± 0.0 | 50.0 ± 0.0 |
| | EGENNET(ours) | 50.0 ± 0.0 | 100.0 ± 0.0 | 100.0 ± 0.0 | 100.0 ± 0.0 | 100.0 ± 0.0 |
| Inv. | Maximally expressiveness I-GGNN | 50% | 50% | 50% | 50% | 50% |
| | SchNet Schütt et al. (2018) | 50.0 ± 0.0 | 50.0 ± 0.0 | 50.0 ± 0.0 | 50.0 ± 0.0 | 50.0 ± 0.0 |
| | DimeNet Gasteiger et al. (2020b) | 50.0 ± 0.0 | 50.0 ± 0.0 | 50.0 ± 0.0 | 50.0 ± 0.0 | 50.0 ± 0.0 |
| | SphereNet Liu et al. (2021) | 50.0 ± 0.0 | 50.0 ± 0.0 | 50.0 ± 0.0 | 50.0 ± 0.0 | 50.0 ± 0.0 |
| | SchNet$_{\text{full graph}}$[§] Schütt et al. (2018) | 100.0 ± 0.0 | 100.0 ± 0.0 | 100.0 ± 0.0 | 100.0 ± 0.0 | 100.0 ± 0.0 |
| | SchNet$_{\text{global feat}}$[§] Schütt et al. (2018) | 100.0 ± 0.0 | 100.0 ± 0.0 | 100.0 ± 0.0 | 100.0 ± 0.0 | 100.0 ± 0.0 |

Table 5: Results on challenging point clouds Pozdnyakov & Ceriotti (2022); Pozdnyakov et al. (2020). The table shows the performance of different models on progressively more difficult tasks (Hard1 to Harder).

| | Point Clouds | Hard1 | Hard2 | Hard3 | Harder |
|---|---|---|---|---|---|
| | GramNet | 1.0 | 1.0 | 1.0 | 1.0 |
| | GeoEGNN | 0.998 | 0.97 | 0.85 | 0.899 |
| | EGNN | 0.5 | 0.5 | 0.5 | 0.5 |
| Results | LinearEGNN | 1.0 | 1.0 | 1.0 | 0.5 |
| | MACE | 1.0 | 1.0 | 1.0 | 1.0 |
| | TFN | 0.5 | 0.5 | 0.55 | 0.5 |
| | DimeNet | 1.0 | 1.0 | 1.0 | 1.0 |
| | GVPGNN | 1.0 | 1.0 | 1.0 | 1.0 |
| | EGENNET (ours) | 1.0 | 1.0 | 1.0 | 1.0 |

by a large margin. In the Drugs 75K, we obtain comparable, but somewhat higher, results than the best method (while in Table 2, in two of three Drugs-75K cases, our results were the best by a small margin).

**Time measurements**   To compare our running time with other models, we measured the execution time for each model on the 12-chain task with all models using 128 hidden feature size. The results 7 show that our running time is on par with the competitors.

**Norm Accuracies**   To heuristically estimate 'how generic' the molecules in the chemical property datasets (Axelrod & Gomez-Bombarelli, 2022; Gensch et al., 2022; Meyer et al., 2018) we considered are, we computed all norms of atoms in each (centralized) molecule and counted the number of unique norms relative to the total number of points. We considered a norm unique if its distance from all other norms exceeded a specified threshold $\mu$, and considered various threshold values. The results 8 indicate that the norms exhibit considerable variation, suggesting that the molecules are close to being generic. This supports the conjecture stated in the main text that the surprising success of our simple method, compared to more expressive methods, is related to the near-genericity of the molecules in these datasets.

Table 6: Raw performance data (mean ± standard deviation) of 1D, 2D, 3D, and conformer ensemble MRL models all taken from Zhu et al. (2024) in terms of absolute test error. Our model is in the 3D category, named EGENNET.

| Category | Model | IP | EA | $\chi$ | $B_5$ | L | BurB$_5$ | BurL | BDE |
|---|---|---|---|---|---|---|---|---|---|
| 1D | Random Forest | 0.4987±0.0037 | 0.4747±0.0022 | 0.2732±0.0031 | 0.4760±0.0041 | 0.4303±0.0090 | 0.2758±0.0180 | 0.1521±0.0149 | 3.03±0.27 |
| 1D | LSTM | 0.4788±0.0024 | 0.4648±0.0002 | 0.2505±0.0050 | 0.4879±0.0280 | 0.5142±0.0411 | 0.2813±0.0041 | 0.1924±0.0028 | 2.82±0.07 |
| 1D | Transformer | 0.6617±0.0023 | 0.5850±0.0031 | 0.4073±0.0006 | 0.9611±0.0813 | 0.8389±0.0431 | 0.4929±0.0369 | 0.2781±0.0207 | 10.08±0.64 |
| 2D | GIN | 0.4354±0.0029 | 0.4169±0.0032 | 0.2260±0.0017 | 0.3128±0.0264 | 0.4003±0.0341 | 0.1719±0.0031 | 0.1200±0.0040 | 2.63±0.22 |
| 2D | GIN-VN | 0.4361±0.0059 | 0.4169±0.0083 | 0.2267±0.0002 | 0.3567±0.0031 | 0.4344±0.0416 | 0.2422±0.0033 | 0.1741±0.0109 | 2.74±0.24 |
| 2D | ChemProp | 0.4595±0.0028 | 0.4417±0.0045 | 0.2441±0.0012 | 0.4850±0.0068 | 0.5452±0.0454 | 0.3002±0.0086 | 0.1948±0.0138 | 2.66±0.14 |
| 2D | GraphGPS | 0.4351±0.0049 | 0.4085±0.0055 | 0.2212±0.0054 | 0.3450±0.0324 | 0.4363±0.0133 | 0.2066±0.0115 | 0.1500±0.0138 | 2.48±0.19 |
| 3D | SchNet | 0.4394±0.0062 | 0.4207±0.0021 | 0.2243±0.0089 | 0.3293±0.0068 | 0.5458±0.0341 | 0.2295±0.0111 | 0.1861±0.0095 | 2.54±0.00 |
| 3D | DimeNet++ | 0.4441±0.0087 | 0.4233±0.0072 | 0.2436±0.0075 | 0.3510±0.0107 | 0.4174±0.0397 | 0.2097±0.0160 | 0.1526±0.0072 | 1.45±0.03 |
| 3D | GemNet | 0.4069±0.0007 | **0.3922±0.0024** | **0.1970±0.0039** | 0.2789±0.0125 | 0.3754±0.0086 | 0.1782±0.0099 | 0.1635±0.0063 | 1.65±0.30 |
| 3D | PaiNN | 0.4505±0.0041 | 0.4495±0.0054 | 0.2324±0.0040 | 0.3443±0.0388 | 0.4471±0.0324 | 0.2395±0.0176 | 0.1673±0.0088 | 2.12±0.09 |
| 3D | ClofNet | 0.4393±0.0084 | 0.4251±0.0066 | 0.2378±0.0020 | 0.4873±0.0093 | 0.6417±0.0362 | 0.2884±0.0166 | 0.2529±0.0052 | 2.60±0.02 |
| 3D | LEFTNet | 0.4174±0.0007 | 0.3964±0.0009 | 0.2083±0.0054 | 0.3072±0.0012 | 0.4493±0.0261 | 0.2176±0.0010 | 0.1486±0.0095 | 1.53±0.05 |
| 3D | EGENNET(Ours) | 0.4251±0.0463 | 0.3935±0.0361 | 0.2332±0.0248 | **0.1953±0.0068** | **0.1956±0.0264** | **0.1545±0.0019** | **0.0644±0.0017** | **0.4455±0.1305** |
| 3D + Sampling | SchNet | 0.4452±0.0080 | 0.4232±0.0042 | 0.2243±0.0022 | 0.3235±0.0147 | 0.4598±0.0041 | 0.2086±0.0111 | 0.1739±0.0142 | 1.97±0.01 |
| 3D + Sampling | DimeNet++ | 0.4395±0.0032 | 0.4217±0.0040 | 0.2432±0.0048 | 0.3323±0.0320 | 0.4153±0.0208 | 0.2237±0.0122 | 0.1561±0.0241 | 1.47±0.03 |
| 3D + Sampling | GemNet | **0.4066±0.0015** | 0.3910±0.0004 | 0.2027±0.0013 | 0.2694±0.0221 | 0.3488±0.0252 | 0.1796±0.0098 | 0.1184±0.0033 | 1.60±0.10 |
| 3D + Sampling | PaiNN | 0.4466±0.0087 | 0.4393±0.0045 | 0.2331±0.0037 | 0.3441±0.0161 | 0.4358±0.0343 | 0.2476±0.0070 | 0.1543±0.0022 | 1.92±0.01 |
| 3D + Sampling | ClofNet | 0.4430±0.0074 | 0.4237±0.0005 | 0.2335±0.0090 | 0.4524±0.0935 | 0.5962±0.0074 | 0.2442±0.0109 | 0.1756±0.0112 | 2.51±0.23 |
| 3D + Sampling | LEFTNet | 0.4149±0.0019 | 0.3988±0.0048 | 0.2141±0.0084 | 0.2834±0.0068 | 0.4407±0.0531 | 0.2120±0.0097 | 0.1547±0.0101 | 1.52±0.00 |
| SchNet | Mean | 0.4583±0.0019 | 0.4410±0.0018 | 0.2371±0.0098 | 0.3075±0.0151 | 0.4691±0.0234 | 0.2282±0.0206 | 0.1619±0.0062 | 2.53±0.02 |
| | DeepSet | 0.4537±0.0065 | 0.4396±0.0010 | 0.2385±0.0066 | 0.3105±0.0381 | 0.4322±0.0464 | 0.2249±0.0234 | 0.1535±0.0076 | 2.29±0.22 |
| | Attention | 0.4556±0.0075 | 0.4382±0.0125 | 0.2380±0.0007 | 0.2704±0.0187 | 0.4517±0.0132 | 0.2024±0.0183 | 0.1443±0.0043 | 2.64±0.00 |
| DimeNet++ | Mean | 0.4488±0.0086 | 0.4340±0.0079 | 0.2425±0.0060 | 0.2630±0.0122 | 0.3828±0.0331 | 0.1960±0.0059 | 0.1268±0.0060 | 1.79±0.12 |
| | DeepSet | 0.4126±0.0076 | 0.3944±0.0034 | 0.2267±0.0047 | 0.2889±0.0069 | 0.3468±0.0090 | 0.1783±0.0110 | 0.1339±0.0087 | 1.75±0.01 |
| | Attention | 0.4188±0.0024 | 0.4030±0.0075 | 0.2325±0.0028 | 0.3718±0.0300 | 0.3628±0.0259 | 0.1899±0.0081 | 0.1185±0.0105 | 2.57±0.21 |
| GemNet | Mean | 0.4505±0.0052 | 0.4334±0.0023 | 0.2289±0.0032 | 0.2635±0.0053 | 0.3753±0.0036 | 0.1671±0.0154 | 0.1587±0.0029 | 2.19±0.06 |
| | DeepSet | 0.4187±0.0022 | 0.4002±0.0012 | 0.2169±0.0036 | 0.2313±0.0026 | 0.3386±0.0269 | 0.1589±0.0068 | 0.0947±0.0012 | 2.25±0.21 |
| | Attention | 0.4212±0.0017 | 0.4221±0.0097 | 0.2260±0.0056 | 0.2670±0.0026 | 0.3554±0.0147 | 0.1769±0.0153 | 0.1346±0.0075 | 2.68±0.02 |
| PaiNN | Mean | 0.4591±0.0024 | 0.4425±0.0064 | 0.2360±0.0032 | 0.2877±0.0252 | 0.3950±0.0233 | 0.1817±0.0091 | 0.1472±0.0039 | 1.87±0.16 |
| | DeepSet | 0.4471±0.0071 | 0.4269±0.0033 | 0.2294±0.0065 | 0.2225±0.0218 | 0.3619±0.0192 | 0.1693±0.0111 | 0.1324±0.0091 | 2.20±0.05 |
| | Attention | 0.4641±0.0016 | 0.4567±0.0094 | 0.2471±0.0049 | 0.3496±0.0140 | 0.4109±0.0167 | 0.2123±0.0005 | 0.1506±0.0029 | 2.23±0.12 |
| ClofNet | Mean | 0.4536±0.0030 | 0.4301±0.0007 | 0.2365±0.0075 | 0.3555±0.0193 | 0.4485±0.0053 | 0.2473±0.0076 | 0.2022±0.0212 | 2.01±0.08 |
| | DeepSet | 0.4280±0.0056 | 0.4033±0.0024 | 0.2199±0.0073 | 0.3228±0.0020 | 0.4742±0.0161 | 0.2263±0.0249 | 0.1548±0.0039 | 2.35±0.04 |
| | Attention | 0.4330±0.0071 | 0.4107±0.0048 | 0.2220±0.0084 | 0.3734±0.0267 | 0.4963±0.0286 | 0.2178±0.0186 | 0.1690±0.0281 | 2.66±0.14 |
| LEFTNet | Mean | 0.4402±0.0062 | 0.4267±0.0026 | 0.2183±0.0007 | 0.2949±0.0001 | 0.3643±0.0352 | 0.2098±0.0146 | 0.1386±0.0007 | 2.04±0.00 |
| | DeepSet | 0.4167±0.0043 | 0.3953±0.0000 | 0.2069±0.0022 | 0.2644±0.0130 | 0.3866±0.0270 | 0.2023±0.0026 | 0.1441±0.0042 | 2.51±0.30 |
| | Attention | 0.4229±0.0059 | 0.4067±0.0047 | 0.2198±0.0011 | 0.3161±0.0116 | 0.4324±0.0292 | 0.2017±0.0023 | 0.1508±0.0075 | 2.63±0.15 |

Table 7: Running times of different methods on the 12-chain problem.

| Method | Time (in secs) |
|---|---|
| SchNet Schütt et al. (2018) | 53.3 |
| DimeNet Gasteiger et al. (2020b) | 222.6 |
| Sphere-Net Liu et al. (2021) | 360.9 |
| EGNN Satorras et al. (2021) | 98.8 |
| GVP Jing et al. (2020) | 222.4 |
| TFN Thomas et al. (2018) | 268.1 |
| MACE Batatia et al. (2022) | 440.6 |
| **EGENNET (ours)** | 112.0 |

Table 8: Norms repetition percentages at various thresholds for each dataset.

| Threshold | Drugs | Kraken | BDE |
|---|---|---|---|
| 0.0 | 1.0 | 1.0 | 1.0 |
| 1e-4 | 0.9967 | 0.9804 | 0.9950 |
| 1e-3 | 0.9678 | 0.9139 | 0.9573 |
| 1e-2 | 0.7230 | 0.5217 | 0.6698 |

