# OpenReview forum: "On the Expressive Power of Sparse Geometric MPNNs"
_ICLR.cc/2025/Conference — ICLR 2025 Poster_

### Official Review · Reviewer_K13L · 2024-11-02

**Soundness:** 2
**Presentation:** 2
**Contribution:** 2
**Rating:** 6
**Confidence:** 3

**Summary:**

This study studies expressivity of message-passing neural networks (MPNNs) for 3D graphs. Unlike earlier research that assuming fully connected graphs, this work demonstrates that MPNNs with equivariant features can distinguish between generic non-isomorphic geometric graphs when the graphs are connected. For cases where only invariant features are used, the ability to differentiate graphs is ensured for those that are generically globally rigid. This work also proposes a new architecture called EGENNET, which utilizes equivariant features and performs well on both synthetic and chemical benchmarks.

**Strengths:**

1. Solid theoretic results. The "generic" describes degeneracies cases well.
2. Separation experiments are clear.

**Weaknesses:**

1. Considering generic geometric graphs only rather than all geometric graphs significantly weakens the significance of theoretic results. Though degenerated cases are rare, they can affect model's performance on generic graphs as neural networks are smooth. The inclusion of high-order irreducible representations solves these cases and boosts performance significantly.[1, 2]

2. Previous work also considers expressivity on geometric graph without assuming fully connection. [3] proves in that in non-degenerated cases, MPNN with equivariant feature can distinguish connected graphs, while MPNN with invariant features only cannot.

3. Experiment should includes graph force field tasks on qm9 and md17 as previous works [2].

[1] Nadav Dym, Haggai Maron. On the Universality of Rotation Equivariant Point Cloud Networks. ICLR 2021.
[2] https://www.nature.com/articles/s41467-022-29939-5
[3] Xiyuan Wang, Muhan Zhang. Graph Neural Network With Local Frame for Molecular Potential Energy Surface. LoG 2022.

**Questions:**

Please refer to weakness.

---

> ### Author Response · Authors · 2024-11-19
> **New version and answers**
>
> **Weakness 1:**
> Considering generic geometric graphs only rather than all geometric graphs significantly weakens the significance of theoretic results. Though degenerated cases are rare, they can affect model's performance on generic graphs as neural networks are smooth. The inclusion of high-order irreducible representations solves these cases and boosts performance significantly.[1, 2]
>
> **Answer 1:**
>
> We agree that the generic assumption is a strong assumption. Nonetheless,  our model demonstrates substantial improvements across three benchmarks in real-world tasks, even in comparison to methods that use high-order irreducible representations like GemNet (see Table 2 in our submission), indicating that our analysis holds practical relevance. We believe this is due to the fact that these datasets do not exhibit many symmetries, as shown by the fact that they do not have many repeated norms. See Table 8 of the revised version of our submission for the details. For more challenging problems with many symmetries, stronger methods may be advisable.
>
> **Weakness 2:**
>
> Previous work also considers expressivity on geometric graph without assuming fully connection. [3] proves in that in non-degenerated cases, MPNN with equivariant feature can distinguish connected graphs, while MPNN with invariant features only cannot.
>
>
> **Answer 2:**
>
> Thanks for suggesting this reference. We have added it to the Related Work section of the paper.
>
> **Weakness 3**
>
> Experiment should includes graph force field tasks on qm9 and md17 as previous works [2].
>
> **Answer 3**
>
>  Thanks for the suggestion. Our paper includes promising results on three molecular property prediction tasks and several synthetic datasets. We think this is sufficient to prove the validity of our results, but we will be happy to look into these applications in future work.

---

> ### Comment · Reviewer_K13L · 2024-11-24
>
> Thank you for the response. My concerns are not solved. For weakness 2, the difference between this work and [3] are not clearly clarified. For weakness 3, these commonly used datasets in previous geometric GNN papers are still not used. Therefore, I will keep my score.

---

> > ### Author Response · Authors · 2024-11-26
> > **Answer to concerns**
> >
> > **W2:**
> > We are happy to expand more on the relationship with [3]:
> >
> > Theorem 4.2 in [3], shows that the proposed equivariant model can identify generic molecules with L message passing iterations, where L is the diameter of the graph. In contrast. In our Theorem 4.1 we only require d+1 iterations (d=3 in standard applications, so only 4 iterations are needed).
> > Theorem 4.2 in [3]  identifies the 3D structure but not the combinatorial structure. Our Theorem 4.1 guarantees recovery of both combinatorial and geometric structure, in the generic case.
> >  Our Theorem 3.1 gives a precise characterization of when invariant models are, and are not, generically complete. This does not appear in the paper you cited.
> >
> > **W3:**
> > We do not think that the fact that these datasets are popular, means that we necessarily have to experiment with them. There are many datasets out there. In fact, in a recent submission, we have experimented with QM9 and were told that the QM9 datasets is widely known in the community to be saturated, or something to that effect. We do not necessarily agree with this view. We are just saying that popularity also has disadvantages.

---

> > > ### Comment · Reviewer_K13L · 2024-11-26
> > >
> > > Thanks. The iteration step difference between yours and [3] is interesting and helps me understand the generic condition better, so I raise my score to 6. Please add the detailed discussion of [3] and more properties of generic condition (like line 964-966) in revision.

---

> > > > ### Author Response · Authors · 2024-11-27
> > > >
> > > > Thanks! We added the discussion of [3] to the related work section. There is a discussion of properties of the generic condition in lines 151-153, if there is anything else you would like us to add about genericity please let us know

---

### Official Review · Reviewer_X9UN · 2024-11-02

**Soundness:** 3
**Presentation:** 2
**Contribution:** 3
**Rating:** 6
**Confidence:** 3

**Summary:**

In this work, the expressive power of in- and equivariant message-passing neural networks (MPNNs) on geometric graphs is studied, where in contrast to prior results that assume full graphs, arbitrary graph topologies are considered. Here, the authors focus on *generic* separation, meaning that the analysis is restricted to graphs without certain symmetries. The analysis reveals that a graph can be generically identified by an invariant GNN iff it is generically globally rigid, and by an equivariant GNN iff it is connected. The authors then introduce a maximally expressive equivariant GNN, called EGenNet, which can generically separate all connected graphs. The expressivity is empirically verified on toy graphs, and significant improvements on benchmarks for chemical property prediction are reported.

**Strengths:**

- Analyzing the expressive power of sparse geometric GNNs is a relevant research direction, closing the gap between theory (where the attention has been mostly on full graphs) and practive (where typically sparse graphs are considered, e.g., via distance cutoffs).
- The characterizations for the expressivity of in- and equivariant GNNs are simple and practical, and are therefore relevant to the research community, in my opinion.
- The empirical results are quite strong, as EGenNet outperforms practically all graph ML methods and geometric GNNs on the Drugs-75k, Kraken, and BDE benchmarks.

**Weaknesses:**

Although I understand that the assumption of genericity makes the analysis tractable, this remains a significant limitation. Real-world geometric graphs, particularly in the chemical domain, can exhibit lots of symmetries that cannot be overlooked.

**Questions:**

- Did you conduct experiments on how EGenNet compares to the other benchmarks on non-generic geometric graphs? A step towards this could be looking at the levels of symmetry that the graphs in the chemical dataset exhibit (i.e., distances between nodes or graph automorphisms), and analyzing the performance of EGenNet vs. the other benchmarks on such graphs.
- In section 5.1, the EGenNet architecture comes a bit out of nowhere. Perhaps, a paragraph with an intuitive explanation why the update step is defined as it is in equation 5 and how it relates to maximal expressivity would add to the overall understanding of the method.
- The results for chemical property prediction could be discussed in more detail. Specifically, a further analysis of the number of parameters or training runtimes of the different models compared in Table 2 would be interesting to see.
- There are lots of presentation issues in the script that should be fixed, e.g., inconsistent usage of `\citep`/`\citet`, misalignment of tables (Table 2, Table 4, Table 5, Table 6), inconsistent caption alignment, Table 6 seeming to be a screenshot from its original source, or "todo" comments (line 451).

---

> ### Author Response · Authors · 2024-11-19
> **New version and midification**
>
> **Weakness:**
> Although I understand that the assumption of genericity makes the analysis tractable, this remains a significant limitation. Real-world geometric graphs, particularly in the chemical domain, can exhibit lots of symmetries that cannot be overlooked.
>
> **Answer:**
> You raise a valid point. Our response is divided into three parts. Firstly, this assumption is essential for distinguishing all connected graphs, as distinguishing all geometric graphs entails separating all graphs, which is a challenging problem with no known polynomial time algorithm (graph isomorphism). Secondly, from a practical perspective, our results on real-world chemical property benchmarks demonstrate substantial improvements across at least four datasets. Thus, the theoretical advantage in the generic setup also benefits practical applications in real-world contexts. Lastly, the fact that the genericity assumption is rather strong highlights the shortcoming of  invariant models which in  some settings fail *even for generic graphs*.
>
> **Question**:
> Did you conduct experiments on how EGenNet compares to the other benchmarks on non-generic geometric graphs? A step towards this could be looking at the levels of symmetry that the graphs in the chemical dataset exhibit (i.e., distances between nodes or graph automorphisms), and analyzing the performance of EGenNet vs. the other benchmarks on such graphs.
>
> **Answer**:
> Thank you for your suggestion. To test this, we have checked the number of repeated norms in the datasets we used in the paper, and found that for the most part these norms are distinct (with a reasonable threshold). We added these results in Table 8 in the revised version.
>
> **Question**:
> In section 5.1, the EGenNet architecture comes a bit out of nowhere. Perhaps, a paragraph with an intuitive explanation why the update step is defined as it is in equation 5 and how it relates to maximal expressivity would add to the overall understanding of the method.
>
> **Answer**:
> Regarding your suggestion, we have incorporated more intuitive explanations of the model in the uploaded version.
>
> **Question**:
> The results for chemical property prediction could be discussed in more detail. Specifically, a further analysis of the number of parameters or training runtimes of the different models compared in Table 2 would be interesting to see.
>
> **Answer**:
> Thanks for the suggestion, In the appendix, we included a table(7) detailing the running times of each model on the 12-chain dataset while all models exhibit a 128-dimensional feature. We show that our model is comparable with the best models and much faster than other models like dimenet and spherenet with same hidden dimension.
>
> **Question**:
>
> There are lots of presentation issues in the script that should be fixed, e.g., inconsistent usage of \citep/\citet, misalignment of tables (Table 2, Table 4, Table 5, Table 6), inconsistent caption alignment, Table 6 seeming to be a screenshot from its original source, or "todo" comments (line 451).
>
> **Answer:**
> All presentation issues have been addressed and corrected in the updated version. Thank you once again for the helpful feedback.

---

> > ### Comment · Reviewer_X9UN · 2024-11-23
> > **Response to Authors**
> >
> > I sincerely thank the authors for their rebuttal and the time they put into addressing my concerns.
> >
> > > Firstly, this assumption is essential for distinguishing all connected graphs, as distinguishing all geometric graphs entails separating all graphs, which is a challenging problem with no known polynomial time algorithm (graph isomorphism).
> >
> > I agree that separating *all* geometric graphs would be unfeasible, but considering that the model should still be able to distinguish *some* types of generic graphs (e.g., if I am not mistaken, trivially all 1-WL distinguishable graphs where adjacent nodes have the same distances), more fine-grained expressivity results would still be interesting (though this is explicitly *not* a demand, but more a suggestion for future work).
> >
> > > To test this, we have checked the number of repeated norms in the datasets we used in the paper
> >
> > I thank the authors for adding this analysis. Table 8 might still benefit from a short explanation how this statistic is calculated.
> >
> > > Regarding your suggestion, we have incorporated more intuitive explanations of the model in the uploaded version.
> >
> > I thank the authors for adding the explanation in lines 329-330, though I still feel that this adds little to the specific intuition that went into designing the update rule in equation 5. I would still suggest the authors to add a more thorough explanation (why do $\phi^{(t, q, 0)}$, $\phi^{(t, q, c)}$ take these specific inputs?).
> >
> > > Thanks for the suggestion, In the appendix, we included a table(7) detailing the running times of each model on the 12-chain dataset while all models exhibit a 128-dimensional feature.
> >
> > I thank the authors for adding this table.
> >
> > > All presentation issues have been addressed and corrected in the updated version.
> >
> > I kindly disagree with the authors that all presentation issues have been addressed.
> > - `\cite`/`\citet` is still used throughout the script, even in places where `\citep` should be used (e.g., L35-36, L45-46, L47-48, ...)
> > - Table captions should be consistently displayed *above* the table and not below (applies to every table except T5).
> > - Table formatting should be consistent (Table 7 and 8 differ from the rest).
> > In these regards, the form still deviates from the official ICLR guidelines.
> >
> > Since my main questions have been addressed and I think that particularly the empirical results are strong, I am raising my score to 6. Nevertheless, I strongly encourage the authors to keep working on the presentation of the paper.

---

> > > ### Author Response · Authors · 2024-11-27
> > >
> > > Thanks for the positive feedback. We have did our best to address the presentation issues you mentioned, as reflected in the current version of the paper. We will continue working on the presentation as you suggest so that the final product is fully polished.

---

### Official Review · Reviewer_BXAA · 2024-11-02

**Soundness:** 2
**Presentation:** 1
**Contribution:** 3
**Rating:** 6
**Confidence:** 3

**Summary:**

The present manuscript  investigates the expressive capabilities of MPNNs for sparse geometric graphs, and it focuses on applications in fields such as chemistry for benchmarking. The authors differentiate between Equivariant Geometric GNNs (E-GGNNs) and Invariant Geometric GNNs (I-GGNNs), claiming that E-GGNNs can separate connected graphs, while I-GGNNs require globally rigid graphs for separation.

They propose EGENNET, a simple E-GGNN model intended to maximize expressiveness in sparse settings, and benchmark its performance against alternative models.

**Strengths:**

1. The paper is clear on addressing the question of expressivity on the specific area of sparse graphs, which is relevant to applications like molecular modeling, where the computational cost of fully connected graphs is prohibitive.

2. The authors provide a theoretical analysis distinguishing the separation power of E-GGNNs and I-GGNNs based on graph connectivity and rigidity, which adds clarity to the strengths and limitations of each approach.

3. The proposed architecture seems to be fairly simple, providing a potentially expressive model that is easier to implement than complex alternatives like (d−1)-WL-based models, aligning with the need for practical applications in sparse graph settings.

**Weaknesses:**

- I find the connection of generic matrices with the use cases of graphs complicated. The authors do not show a clear motivation on defining expressivity over generic spaces for the use over graphs. Can the authors motivate better the analysis over generic spaces? What happens in the non-genericity direction?

- The tradeoff between efficiency and expressiveness is confusing. It seems that still if we want to achieve maximal expressiveness leads to quadratic complexity (G.2 Section), not having a clear understanding of the expressivity loss, once we set C < 2Nd + 1.
- The presentation is very poor. The theoretical analysis is hard to follow. There are a lot of typos, and inconsistencies. There are forgotten TODOs in the main text (check Section 6.2), and elements of the paper are not even visible (for example Table 5 does not fit the page).

Overall, the paper shows some elements of haste, and I would suggest the improvement of the writing and the presentation of the theoretical as well as the empirical results.

**Questions:**

Given the reliance of the paper's results on the genericity assumption, can the authors motivate clearly the need to study the genericity case? What can we say about non-generic spaces, and why are they important to their experimental results? e.g. for the molecular property prediction tasks.


UPDATE: Given the answer by the authors and their efforts to fix the presentation issues of the paper, I'm raising my rating.

---

> ### Author Response · Authors · 2024-11-19
> **Our answer to your requests**
>
> **Weakness:**
> I find the connection of generic matrices with the use cases of graphs complicated. The authors do not show a clear motivation on defining expressivity over generic spaces for the use over graphs. Can the authors motivate better the analysis over generic spaces? What happens in the non-genericity direction?
>
> **Answer:**
> We believe these issues are discussed extensively in the introduction, mostly in the paragraphs leading to Question 1. We will summarize the discussion: geometric graphs are graphs which have continuous 3D node features. They arise in many applications, e.g., as models for molecules. GNNs which are complete when considering all geometric graphs, discussed in [1],[2] require a unrealisticially high computational complexity of n^4. This motivates the problem of considering graphs with ‘generic’ 3D coordinates: almost all graphs, in the Lebesgue sense, are generic, and when restricting to such graphs simple models with linear to quadradic complexity (see next question) are sufficient, as we show here.
>
> **Weakness:**
> The tradeoff between efficiency and expressiveness is confusing. It seems that still if we want to achieve maximal expressiveness leads to quadratic complexity (G.2 Section), not having a clear understanding of the expressivity loss, once we set \( C < 2Nd + 1 \).
>
>
> **Answer:**
> You are right. According to the analysis presented in the paper, the worst-case complexity is indeed quadratic. This is still much more efficient than   provably complete models which have n^4 complexity, as described in [1].
>
>  We would also like to point out that mathematical complexity and approximation bounds are often unrealistically high. As researchers in this field for several years, we actually regard both the n^4 complexity in [1] and the n^2 complexity here as significantly good bounds. For comparison of much higher bounds for arguably  simpler problems see for example [3].
>
> Finally, we note that our analysis could easily be extended to show that, under the common ‘manifold hypothesis’, which states that the data comes from a low k dimensional manifold, only C=O(k) channels will suffice for generic completeness. In this case, the complexity would be truly linear. We have avoided going into these details in the paper, since, as the reviewers have noted, this paper is already mathematically involved.
>
> **Weakness:**
> The theoretical analysis is hard to follow. There are a lot of typos, and inconsistencies. There are forgotten TODOs in the main text (check Section 6.2), and elements of the paper are not even visible (for example Table 5 does not fit the page).
>
>
> **Answer:**
> Regarding typos and other writing issues, we have updated them. If the theoretical analysis is difficult to follow in certain sections, please let us know which parts are challenging, and we will clarify them accordingly.
>
>
>
> [1] Hordan, Snir, et al. "Complete neural networks for Euclidean graphs." AAAI 2024),
>
>  [2]- Delle Rose, Valentino, et al. "Three iterations of (d−1)-WL test distinguish non-isometric clouds of d-dimensional points." Advances in Neural Information Processing Systems 36 (2024)
>
> [3] Polynomial Width is Sufficient for Set Representation with High Dimensional Features. Wang et al. ICLR 2024

---

> > ### Comment · Reviewer_BXAA · 2024-11-26
> >
> > Thank you for answering my questions. Your answers on both questions about generic spaces, and computational complexity are clear and address my concern. Thus, I'm going to raise my rating.

---

### Official Review · Reviewer_CwLH · 2024-11-04

**Soundness:** 3
**Presentation:** 3
**Contribution:** 3
**Rating:** 6
**Confidence:** 3

**Summary:**

The paper proposes a method for distinguishing between geometric graphs based on the connectivity of the graph structure. Moreover they propose an equivariant architecture which they claim to be maximal in expressivity.

**Strengths:**

The paper utilizes global rigidity to develop a novel direction for developing geometric graph neural networks.

 EGenNet achieves state of the art performance and makes significant improvements on a few datasets.

**Weaknesses:**

I have two major concerns regarding the submitted work.

1) Incomplete Submission: The paper contains 'TODO' placeholders and formatting issues, indicating it is an unfinished draft. This severely hinders the ability to conduct a thorough and fair review and compromises the integrity of the review process and the conference's high standards.

2) Lack of Consistency and Discussion in Empirical Results: The empirical results lack completeness and coherence. Key sections are marked TODO, and there are inconsistencies between tables, making it difficult to interpret the significance of the findings. A more thorough discussion and clear presentation of results are necessary to support the paper's claims.

**Questions:**

Below is a partial list of incomplete work:

1) The paper is not formatted to the ICLR standard, e.g. it is missing the header **Under review as a conference paper at ICLR 2025**.

2) The inline citation format is incorrect.

3) Section 6.2 contains a "TODO" placeholder.

4) The second paragraph of Section 5 is two egregiously run on sentences.

5) Tables 2, 4, 5, and 6 extend beyond the page margins, with Table 5 exceeding the page to the point where the results are unreadable.

6) Table 4 uses an incorrect title for the model, referring to it as GenNet.

7) Table 6 is a low resolution image of a table and has discrepancies in the reported results by comparison to Table 2.

---

1) The definition of "identifies" appears to be equivalent to the definition of a complete invariant function F. Per Section 1 and Appendix B: "The main text discussed (sic) how generic completeness can be achieved using I-GGNN when the graph is globally rigid." Is there a reason that the terminology of completeness that appears in the introduction and appendix is dropped in the definitions of Section 2 in favor of identifies? Maintaining consistent terminology would significantly enhance clarity.

2) The cutoff tables 5 and inconsistencies between Table 6 and Table 3 make it difficult to interpret the significance of the reported results. Could the authors please explain the discrepancies between the Tables.

3) In Kraken BurL and BDE it appears that EGenNet significantly outperforms the previous state of the art, especially for BDE. Could the authors provide an explanation for why this major improvement occurs for the BDE dataset?

---

> ### Author Response · Authors · 2024-11-19
> **Our answer to all your requests**
>
> All our modifications were added in blue.
>
> **Weaknesses 1:**
> I have two major concerns regarding the submitted work.
> Incomplete Submission: The paper contains 'TODO' placeholders and formatting issues, indicating it is an unfinished draft. This severely hinders the ability to conduct a thorough and fair review and compromises the integrity of the review process and the conference's high standards. Additionally, these issues distract from the paper’s overall clarity and presentation, which could lead to misunderstandings of the proposed ideas and results.
>
> **Answer 1:**
> You are absolutely correct, the paper did indeed contain a (single) TODO placeholder and had some formatting issue. We apologize for the oversight. In the new version we uploaded, all of these issues have been addressed as per your feedback. Aside from these issues, we believe the paper is well-written and in solid form. In fact, we have finished this paper early, had carefully rewritten it several times, and have received feedback from several peers regarding our work. The missed TODO remark was aimed to fix one of these remarks. We again apologize for the oversight but would like to stress that a lot of work was put into this submission, and it is not an “unfinished draft”.
>
> **Weakness 2**: Lack of Consistency and Discussion in Empirical Results:
> The empirical results lack completeness and coherence… A more thorough discussion and clear presentation of results are necessary to support the paper's claims.
>
> **Answer 2:**
> Thank you for your remark. In the new submission, the empirical results section has been updated, with the objectives and findings explicitly explained for each section of experiments; please take a look. We have added our objectives for the experiments and clearly explained the results obtained.
>
> **Questions:**
> Below is a partial list of incomplete work:
> The paper is not formatted to the ICLR standard, e.g., it is missing the header "Under review as a conference paper at ICLR 2025."
> The inline citation format is incorrect.
> Section 6.2 contains a "TODO" placeholder.
> The second paragraph of Section 5 contains two egregiously run-on sentences.
> Tables 2, 4, 5, and 6 extend beyond the page margins, with Table 5 exceeding the page to the point where the results are unreadable.
> Table 4 uses an incorrect title for the model, referring to it as GenNet.
>
> **Answer :**
> All table, template, and formatting issues have been resolved in the updated version. Thanks again.
>
> **Questions (theoretical):**
> The definition of "identifies" appears to be equivalent to the definition of a complete invariant function F. Per Section 1 and Appendix B: "The main text discussed how generic completeness can be achieved using I-GGNN when the graph is globally rigid." Is there a reason that the terminology of completeness that appears in the introduction and appendix is dropped in the definitions of Section 2 in favor of "identifies"? Maintaining consistent terminology would significantly enhance clarity
>
> **Answer:**
> The term ‘complete’ is a somewhat informal term used in the literature. We do not define it here. The term identifies has a precise mathematical definition. Roughly, though, a model is complete if it identifies all graphs. We added a comment to this affect in the new version of the paper (page 4)
>
>
> **Question:**
> The cutoff in Table 5 and inconsistencies between Table 6 and Table 3 make it difficult to interpret the significance of the reported results. Could the authors please explain the discrepancies between the tables
>
> **Answer:**
> Regarding the tables, I believe you are referring to Tables 2 and 6. In Table 2, we show the best seed, while in Table 6, we present the average overall seeds. The same two tables were presented in the paper introducing this dataset (Zhu et al. 2024).
>
> **Question:**
> In Kraken BurL and BDE, it appears that EGenNet significantly outperforms the previous state of the art, especially for BDE. Could the authors provide an explanation for why this major improvement occurs for the BDE dataset?
> **.
>
> **Answer:**
> We conjecture this is because these datasets are ‘close to being generic’. To check this conjecture, we
>  checked the norms of atoms in the centralized molecules of the dataset to see if they were repetitive or not. We found that the molecules contain a very small percentage of repeated norms, confirming our conjecture.  These results were added to our revised submission in Table 8.

---

> > ### Comment · Reviewer_CwLH · 2024-11-23
> > **Response to Authors**
> >
> > Thank you for the detailed feedback and significant revisions that have improved the work. I would like to re-emphasize the importance of taking additional time to review and revise. Please note Krakrn in Table 2 and dimenet spherenet in Table 7.
> >
> > One suggested improvement for the norm study is to consider the thresholds relative to the known error tolerance for the positions.
> >
> > I find the work compelling and have raised my score.

---

> > > ### Author Response · Authors · 2024-11-27
> > >
> > > Thanks for the positive feedback, and for the suggested corrections! We have updated the paper, fixing the typos you found, \citet vs. \citep issues, and some other minor typos we found.

---

### Author Response · Authors · 2024-11-19
**New version**

Dear reviewers, thank you for your feedback.

We have uploaded a new version of the paper incorporating the changes you suggested. Changes are colored in blue. In particular, we have added an appendix at the end of the paper with (a) timing of the methods used in the paper and (b) an experiment counting the number of unique norms in our datasets. This experiment show that most norms in our datasets are unique, and hence the data we work with in this paper is ‘close to generic’.

---

### Meta-Review · Area_Chair_UEEm · 2024-12-20

**Metareview:**

This paper studies the expressive power of different GNNs for sparse geometric graphs. The reviewers valued the theoretical contribution as solid and appreciated the appropriateness of the numerical experiments.

During the initial round of reviews, the reviewers raised major concerns about the paper's presentation. It seems that it was hastily written since it didn't comply with the ICLR format and had some TODO placeholders. However, after the discussion period and revision, all negative reviews were raised to 6.

They also raised technical concerns, but they were satisfied with the authors' response.

**Additional Comments On Reviewer Discussion:**

After the discussion period and revision, all negative reviews were raised to 6.

---

### Decision · Program_Chairs · 2025-01-22

Accept (Poster)